# The Effect of Intrinsic Dataset Properties on Generalization: Unraveling Learning Differences Between Natural and Medical Images

**Nicholas Konz**[1]**, Maciej A. Mazurowski**[1,2,3,4]
[1] Department of Electrical and Computer Engineering, [2] Department of Radiology,
[3] Department of Computer Science, [4] Department of Biostatistics & Bioinformatics
Duke University, NC, USA
{nicholas.konz, maciej.mazurowski}@duke.edu

## Abstract

This paper investigates discrepancies in how neural networks learn from different imaging domains, which are commonly overlooked when adopting computer vision techniques from the domain of natural images to other specialized domains such as medical images. Recent works have found that the generalization error of a trained network typically increases with the intrinsic dimension ($d_{\mathrm{data}}$) of its training set. Yet, the steepness of this relationship varies significantly between medical (radiological) and natural imaging domains, with no existing theoretical explanation. We address this gap in knowledge by establishing and empirically validating a generalization scaling law with respect to $d_{\mathrm{data}}$, and propose that the substantial scaling discrepancy between the two considered domains may be at least partially attributed to the higher intrinsic "label sharpness" ($K_{\mathcal{F}}$) of medical imaging datasets, a metric which we propose. Next, we demonstrate an additional benefit of measuring the label sharpness of a training set: it is negatively correlated with the trained model's adversarial robustness, which notably leads to models for medical images having a substantially higher vulnerability to adversarial attack. Finally, we extend our $d_{\mathrm{data}}$ formalism to the related metric of learned representation intrinsic dimension ($d_{\mathrm{repr}}$), derive a generalization scaling law with respect to $d_{\mathrm{repr}}$, and show that $d_{\mathrm{data}}$ serves as an upper bound for $d_{\mathrm{repr}}$. Our theoretical results are supported by thorough experiments with six models and eleven natural and medical imaging datasets over a range of training set sizes. Our findings offer insights into the influence of intrinsic dataset properties on generalization, representation learning, and robustness in deep neural networks.[1]

## 1 Introduction

There has been recent attention towards how a neural network's ability to generalize to test data relates to the *intrinsic dimension* $d_{\mathrm{data}}$ of its training dataset, *i.e.*, the dataset's inherent "complexity" or the minimum degrees of freedom needed to represent it without substantial information loss (Gong et al., 2019). Recent works have found that generalization error typically increases with $d_{\mathrm{data}}$, empirically (Pope et al., 2020) or theoretically (Bahri et al., 2021). Such "scaling laws" with respect to intrinsic dataset properties are attractive because they may describe neural network behavior in generality, for different models and/or datasets, allowing for better understanding and predictability of the behavior, capabilities, and challenges of deep learning. However, a recent study (Konz et al., 2022) showed that generalization scaling behavior differs drastically depending on the input image type, *e.g.*, natural or medical images, showing the non-universality of the scaling law and motivating us to consider its relationship to properties of the dataset and imaging domain.[2]

In this work, we provide theoretical and empirical findings on how measurable intrinsic properties of an image dataset can affect the behavior of a neural network trained on it. We show that certain

---

[1]Code link: `https://github.com/mazurowski-lab/intrinsic-properties`.
[2]Here we take "medical" images to refer to radiology images (*e.g.*, x-ray, MRI), the most common type.

dataset properties that differ between imaging domains can lead to discrepancies in behaviors such as generalization ability and adversarial robustness. Our contributions are summarized as follows.

First, we introduce the novel measure of the intrinsic **label sharpness** ($K_\mathcal{F}$) of a dataset (defined in Section 3.2). The label sharpness essentially measures how similar images in the dataset can be to each other while still having different labels, and we find that it usually differs noticeably between natural and medical image datasets. We then derive and test a neural network generalization scaling law with respect to dataset intrinsic dimension $d_{\text{data}}$, which includes $K_\mathcal{F}$. Our experiments support the derived scaling behavior within each of these two domains, and show a distinct difference in the scaling rate between them. According to our scaling law and likelihood analysis of observed generalization data (Appendix C.1), this may be due to the measured $K_\mathcal{F}$ being typically higher for medical datasets.

Next, we show how a model's adversarial robustness relates to its training set's $K_\mathcal{F}$, and show that over a range of attacks, robustness decreases with higher $K_\mathcal{F}$. Indeed, medical image datasets, which have higher $K_\mathcal{F}$, **are typically more susceptible to adversarial attack than natural image datasets**. Finally, we extend our $d_{\text{data}}$ formalism to derive and test a generalization scaling law with respect to the intrinsic dimension of the model's *learned representations*, $d_{\text{repr}}$, and reconcile the $d_{\text{data}}$ and $d_{\text{repr}}$ scaling laws to show that $d_{\text{data}}$ serves as an approximate upper bound for $d_{\text{repr}}$. We also provide many additional results in the supplementary material, such as a likelihood analysis of our proposed scaling law given observed generalization data (Appendix C.1), the evaluation of a new dataset in a third domain (Appendix C.2), an example of a practical application of our findings (Appendix C.3), and more.

All theoretical results are validated with thorough experiments on six CNN architectures and eleven datasets from natural and medical imaging domains over a range of training set sizes. We hope that our work initiates further study into how network behavior differs between imaging domains.

## 2 RELATED WORKS

We are interested in the scaling of the generalization ability of supervised convolutional neural networks with respect to intrinsic properties of the training set. Other works have also explored generalization scaling with respect to parameter count or training set size for vision or other modalities (Caballero et al., 2023; Kaplan et al., 2020; Hoffmann et al., 2022; Touvron et al., 2023). Note that we model the intrinsic dimension to be constant throughout the dataset's manifold as in Pope et al. (2020); Bahri et al. (2021) for simplicity, as opposed to the recent work of Brown et al. (2023), which we find to be suitable for interpretable scaling laws and dataset properties.

Similar to *dataset* intrinsic dimension scaling (Pope et al., 2020; Bahri et al., 2021; Konz et al., 2022), recent works have also found a monotonic relationship between a network's generalization error and the intrinsic dimension of both the learned hidden layer representations (Ansuini et al., 2019), or some measure of intrinsic dimensionality of the trained model itself (Birdal et al., 2021; Andreeva et al., 2023). In this work, we focus on the former, as the latter model dimensionality measures are typically completely different mathematical objects than the intrinsic dimension of the manifolds of data or representations. Similarly, Kvinge et al. (2023) found a correlation between prompt perplexity and representation intrinsic dimension in Stable Diffusion models.

## 3 PRELIMINARIES

We consider a binary classification dataset $\mathcal{D}$ of points $x \in \mathbb{R}^n$ with target labels $y = \mathcal{F}(x)$ defined by some unknown function $\mathcal{F} : \mathbb{R}^n \to \{0, 1\}$, split into a training set $\mathcal{D}_{\text{train}}$ of size $N$ and test set $\mathcal{D}_{\text{test}}$. The manifold hypothesis (Fefferman et al., 2016) assumes that the input data $x$ lies approximately on some $d_{\text{data}}$-dimensional manifold $\mathcal{M}_{d_{\text{data}}} \subset \mathbb{R}^n$, with $d_{\text{data}} \ll n$. More technically, $\mathcal{M}_{d_{\text{data}}}$ is a metric space such that for all $x \in \mathcal{M}_{d_{\text{data}}}$, there exists some neighborhood $U_x$ of $x$ such that $U_x$ is homeomorphic to $\mathbb{R}^{d_{\text{data}}}$, defined by the standard $L_2$ distance metric $||\cdot||$.

As in Bahri et al. (2021), we consider over-parameterized (number of parameters $\gg N$) models $f(x) : \mathbb{R}^n \to \{0, 1\}$, that are "well-trained" and learn to interpolate all training data: $f(x) = \mathcal{F}(x)$ for all $x \in \mathcal{D}_{\text{train}}$. We use a non-negative loss function $L$, such that $L = 0$ when $f(x) = \mathcal{F}(x)$. Note that we write $L$ as the expected loss over a set of test set points. We assume that $\mathcal{F}$, $f$ and

$L$ are Lipschitz/smooth on $\mathcal{M}_{d_{\text{data}}}$ with respective constants $K_{\mathcal{F}}$, $K_f$ and $K_L$. Note that we use the term "*Lipschitz constant*" of a function to refer to the smallest value that satisfies the Lipschitz inequality.[3] We focus on binary classification as in Pope et al. (2020); Konz et al. (2022), but we note that our results extend naturally to the multi-class case (see Appendix A.1 for more details).

## 3.1 ESTIMATING DATASET INTRINSIC DIMENSION

Here we introduce two common intrinsic dimension estimators for high-dimensional datasets that we use in our experiments, which have been used in prior works on image datasets (Pope et al., 2020; Konz et al., 2022) and learned representations (Ansuini et al., 2019; Gong et al., 2019).

**MLE:** The MLE (maximum likelihood estimation) intrinsic dimension estimator (Levina & Bickel, 2004; MacKay & Ghahramani, 2005) works by assuming that the number of datapoints enclosed within some $\epsilon$-ball about some point on $\mathcal{M}_{d_{\text{data}}}$ scales not as $\mathcal{O}(\epsilon^n)$, but $\mathcal{O}(\epsilon^{d_{\text{data}}})$, and then solving for $d_{\text{data}}$ with MLE after modeling the data as sampled from a Poisson process. This results in $\hat{d}_{\text{data}} = \left[ \frac{1}{N(k-1)} \sum_{i=1}^{N} \sum_{j=1}^{k-1} \log \frac{T_k(x_i)}{T_j(x_i)} \right]^{-1}$, where $T_j(x)$ is the $L_2$ distance from $x$ to its $j^{th}$ nearest neighbor and $k$ is a hyperparameter; we set $k = 20$ as in Pope et al. (2020); Konz et al. (2022). **TwoNN:** TwoNN (Facco et al., 2017) is a similar approach that instead relies on the ratio of the first- and second-nearest neighbor distances. We default to using the MLE method for $d_{\text{data}}$ estimation as Pope et al. (2020) found it to be more reliable for image data than TwoNN, but we still evaluate with TwoNN for all experiments. Note that these estimators do not use datapoint labels.

## 3.2 ESTIMATING DATASET LABEL SHARPNESS

Another property of interest is an empirical estimate for the "label sharpness" of a dataset, $K_{\mathcal{F}}$. This measures the extent to which images in the dataset can resemble each other while still having different labels. Formally, $K_{\mathcal{F}}$ is the Lipschitz constant of the ground truth labeling function $\mathcal{F}$, *i.e.*, the smallest positive $K_{\mathcal{F}}$ that satisfies $K_{\mathcal{F}}||x_1 - x_2|| \geq |\mathcal{F}(x_1) - \mathcal{F}(x_2)| = |y_1 - y_2|$ for all $x_1, x_2 \sim \mathcal{M}_{d_{\text{data}}}$, where $y_i = \mathcal{F}(x_i) \in \{0, 1\}$ is the target label for $x_i$. We estimate this as

$$\hat{K}_{\mathcal{F}} := \max_{j,k} \left( \frac{|y_j - y_k|}{||x_j - x_k||} \right), \tag{1}$$

computed over all $M^2$ pairings $((x_j, y_j), (x_k, y_k))$ of some $M$ evenly class-balanced random samples $\{(x_i, y_i)\}_{i=1}^{M}$ from the dataset $\mathcal{D}$. We use $M = 1000$ in practice, which we found more than sufficient for a converging estimate, and it takes <1 sec. to compute $\hat{K}_{\mathcal{F}}$. We minimize the effect of trivial dataset-specific factors on $\hat{K}_{\mathcal{F}}$ by linearly normalizing all images to the same range (Sec. 4), and we note that both $\hat{K}_{\mathcal{F}}$ and $d_{\text{data}}$ are invariant to image resolution and channel count (Appendix B.1). As the natural image datasets have multiple possible combinations of classes for the binary classification task, we report $\hat{K}_{\mathcal{F}}$ averaged over 25 runs of randomly chosen class pairings.

## 4 DATASETS, MODELS AND TRAINING

**Medical Image Datasets.** We conducted our experiments on seven public medical image (radiology) datasets from diverse modalities and anatomies for different binary classification tasks. These are (1) brain MRI glioma detection (**BraTS**, Menze et al. (2014)); (2) breast MRI cancer detection (**DBC**, Saha et al. (2018)); (3) prostate MRI cancer risk scoring (**Prostate MRI**, Sonn et al. (2013)); (4) brain CT hemorrhage detection (**RSNA-IH-CT**, Flanders et al. (2020)); (5) chest X-ray pleural effusion detection (**CheXpert**, Irvin et al. (2019)); (6) musculoskeletal X-ray abnormality detection (**MURA**, Rajpurkar et al. (2017)); and (7) knee X-ray osteoarthritis detection (**OAI**, Tiulpin et al. (2018)). All dataset preparation and task definition details are provided in Appendix G.

**Natural Image Datasets.** We also perform our experiments using four common "natural" image classification datasets: **ImageNet** (Deng et al., 2009), **CIFAR10** (Krizhevsky et al., 2009), **SVHN** (Netzer et al., 2011), and **MNIST** (Deng, 2012).

---

[3]A subtlety here is that our Lipschitz assumptions only involve pairs of datapoints sampled from the true data manifold $\mathcal{M}_{d_{\text{data}}}$; adversarially-perturbed images (Goodfellow et al., 2015) are not included.

For each dataset, we create training sets of size $N \in \{500, 750, 1000, 1250, 1500, 1750\}$, along with a test set of 750 examples. These splits are randomly sampled with even class-balancing from their respective base datasets. For the natural image datasets we choose two random classes (different for each experiment) to define the binary classification task, and all results are averaged over five runs using different class pairs.[4] Images are resized to $224 \times 224$ and normalized linearly to $[0, 1]$.

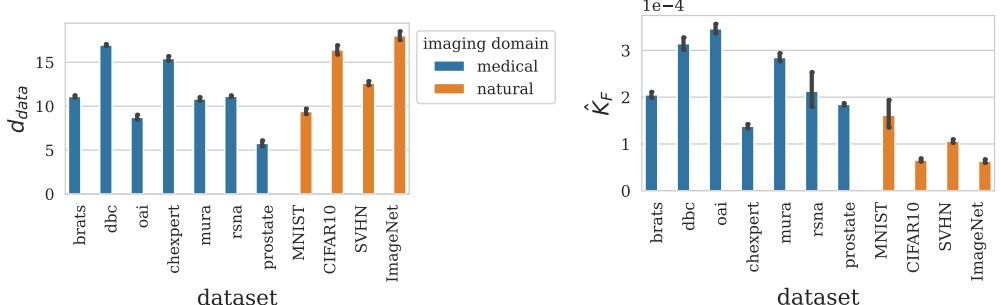

Figure 1: Measured intrinsic dimension ($d_{\text{data}}$, **left**) and label sharpnesses ($\hat{K}_{\mathcal{F}}$, **right**) of the natural (orange) and medical (blue) image datasets which we analyze (Sec. 4). $\hat{K}_{\mathcal{F}}$ **is typically higher for the medical datasets**. $d_{\text{data}}$ values are averaged over all training set sizes, and $\hat{K}_{\mathcal{F}}$ over all class pairings (Sec. 3.2); error bars indicate $95\%$ confidence intervals.

**Models and training.** We evaluate six models total: ResNet-18, -34 and -50 (He et al., 2016), and VGG-13, -16 and -19 (Simonyan & Zisserman, 2015). Each model $f$ is trained on each dataset for its respective binary classification task with Adam (Kingma & Ba, 2015) until the model fully fits to the training set, for each training set size $N$ described previously. We provide all training and implementation details in Appendix F, and our code can be found at https://github.com/mazurowski-lab/intrinsic-properties.

# 5 THE RELATIONSHIP OF GENERALIZATION WITH DATASET INTRINSIC DIMENSION AND LABEL SHARPNESS

In Fig. 1 we show the average measured intrinsic dimension $d_{\text{data}}$ and label sharpness $\hat{K}_{\mathcal{F}}$ of each dataset we study. While both natural and medical datasets can range in $d_{\text{data}}$, we note that medical datasets typically have much higher $\hat{K}_{\mathcal{F}}$ than natural image datasets, which we will propose may explain differences in generalization ability scaling rates between the two imaging domains. We emphasize that $d_{\text{data}}$ and $K_{\mathcal{F}}$ are model-independent properties of a dataset itself. We will now describe how network generalization ability scales with $d_{\text{data}}$ and $K_{\mathcal{F}}$.

## 5.1 BOUNDING GENERALIZATION ABILITY WITH DATASET INTRINSIC DIMENSION

A result which we will use throughout is that on average, given some $N$ datapoints sampled i.i.d. from a $d$-dimensional manifold, the distance between the nearest neighbor $\hat{x}$ of some datapoint $x$ scales as $\mathbb{E}_x \|x - \hat{x}\| = \mathcal{O}(N^{-1/d_{\text{data}}})$ (Levina & Bickel, 2004). As such, the nearest-neighbor distance of some test point to the training set decreases as the training set grows larger by $\mathcal{O}(N^{-1/d_{\text{data}}})$. It can then be shown that the loss on the test set/generalization error scales as $\mathcal{O}(K_L \max(K_f, K_{\mathcal{F}})N^{-1/d_{\text{data}}})$ on average; this is summarized in the following theorem.

**Theorem 1** (Generalization Error and Dataset Intrinsic Dim. Scaling Law (Bahri et al., 2021)). *Let $L, f$ and $\mathcal{F}$ be Lipschitz on $\mathcal{M}_{d_{\text{data}}}$ with respective constants $K_L, K_f$ and $K_{\mathcal{F}}$. Further let $\mathcal{D}_{\text{train}}$ be a training set of size $N$ sampled i.i.d. from $\mathcal{M}_{d_{\text{data}}}$, with $f(x) = \mathcal{F}(x)$ for all $x \in \mathcal{D}_{\text{train}}$. Then, $L = \mathcal{O}(K_L \max(K_f, K_{\mathcal{F}})N^{-1/d_{\text{data}}})$.*

---

[4]$N = 1750$ is the upper limit of $N$ that all datasets could satisfy, given the smaller size of medical image datasets and ImageNet's typical example count per class. In Appendix C.4 we evaluate much higher $N$ for datasets that allow for it.

We note that the $K_{\mathcal{F}}$ term is typically treated as an unknown constant in the literature (Bahri et al., 2021); instead, we propose to *estimate* it with the empirical label sharpness $\hat{K}_{\mathcal{F}}$ (Sec. 3.2). We will next show that $K_f \simeq K_{\mathcal{F}}$ for large $N$ (common for deep models), which allows us to approximate Theorem 1 as $L \simeq \mathcal{O}(K_L K_{\mathcal{F}} N^{-1/d_{\text{data}}})$, **a scaling law independent of the trained model** $f$. Intuitively, this means that the Lipschitz smoothness of $f$ molds to the smoothness of the label distribution as the training set grows larger and test points typically become closer to training points.

**Theorem 2** (Approximating $K_f$ with $K_{\mathcal{F}}$). *$K_f$ converges to $K_{\mathcal{F}}$ in probability as $N \to \infty$.*

We show the full proof in Appendix A.2 due to space constraints. This result is also desirable because computing an estimate for $K_f$, the Lipschitz constant of the model $f$, either using Eq. (1) or with other techniques (Fazlyab et al., 2019), depends on the choice of model $f$, and may require many forward passes. Estimating $K_{\mathcal{F}}$ (Eq. (1) is far more tractable, as it is an intrinsic property of the dataset itself which is relatively fast to compute.

Next, note that the Lipschitz constant $K_L$ is a property of the loss function, which we take as fixed *a priori*, and so does not vary between datasets or models. As such, $K_L$ can be factored out of the scaling law of interest, such that we can simply consider $L \simeq \mathcal{O}(K_{\mathcal{F}} N^{-1/d_{\text{data}}})$, *i.e.*,

$$\log L \lesssim -\frac{1}{d_{\text{data}}} \log N + \log K_{\mathcal{F}} + a \tag{2}$$

for some constant $a$. In the following section, we will demonstrate how the prediction of Eq. (2) may explain recent empirical results in the literature where the rate of this generalization scaling law differed drastically between natural and medical datasets, via the measured differences in the typical label sharpness $\hat{K}_{\mathcal{F}}$ of datasets in these two domains.

## 5.2 Generalization Discrepancies Between Imaging Domains

Consider the result from Eq. (2) that the test loss/generalization error scales approximately as $L \propto K_{\mathcal{F}} N^{-1/d_{\text{data}}}$ on average. From this, we hypothesize that a higher label sharpness $K_{\mathcal{F}}$ will result in the test loss curve that grows faster with respect to $d_{\text{data}}$.

In Fig. 2 we evaluate the generalization error (log test loss) scaling of all models trained on each natural and medical image dataset with respect to the training set intrinsic dimension $d_{\text{data}}$, for all evaluated training set sizes $N$. We also show the scaling of test *accuracy* in Appendix E.1.

We see that *within* an imaging domain (natural or medical), model generalization error typically increases with $d_{\text{data}}$, as predicted, similar to prior results (Pope et al., 2020; Konz et al., 2022); in particular, approximately $\log L \propto -1/d_{\text{data}} + \text{const.}$, aligning with Eq. (2). However, we also see that the generalization error scaling is much sharper for models trained on medical data than natural data; models trained on datasets with similar $d_{\text{data}}$ and of the same size $N$ tend to perform much worse if the data is medical images. A similarly large gap appears for the scaling of test accuracy (Appendix E.1). We posit that this difference is explained by medical datasets typically having much higher label sharpness ($\hat{K}_{\mathcal{F}} \sim 2.5 \times 10^{-4}$) than natural images ($\hat{K}_{\mathcal{F}} \sim 1 \times 10^{-4}$) (Fig. 1) , as $K_{\mathcal{F}}$ is the only term in Eq. (2) that differs between two models with the same training set intrinsic dimension $d_{\text{data}}$ and size $N$. Moreover, in Appendix C.1 we show that accounting for $K_{\mathcal{F}}$ increases the likelihood of the posited scaling law given the observed generalization data. However, we note that there could certainly be other factors causing the discrepancy which are not accounted for.

Intuitively, the difference in dataset label sharpness $K_{\mathcal{F}}$ between these imaging domains is reasonable, as $K_{\mathcal{F}}$ describes how similar a dataset's images can be while still having different labels (Sec. 3.2). For natural image classification, images from different classes are typically quite visually distinct. However, in many medical imaging tasks, a change in class can be due to a small change or abnormality in the image, resulting in a higher dataset $K_{\mathcal{F}}$; for example, the presence of a small breast tumor will change the label of a breast MRI from healthy to cancer.

## 6 Adversarial Robustness and Training Set Label Sharpness

In this section we present another advantage of obtaining the sharpness of the dataset label distribution ($K_{\mathcal{F}}$): it is negatively correlated with the adversarial robustness of a neural network. Given

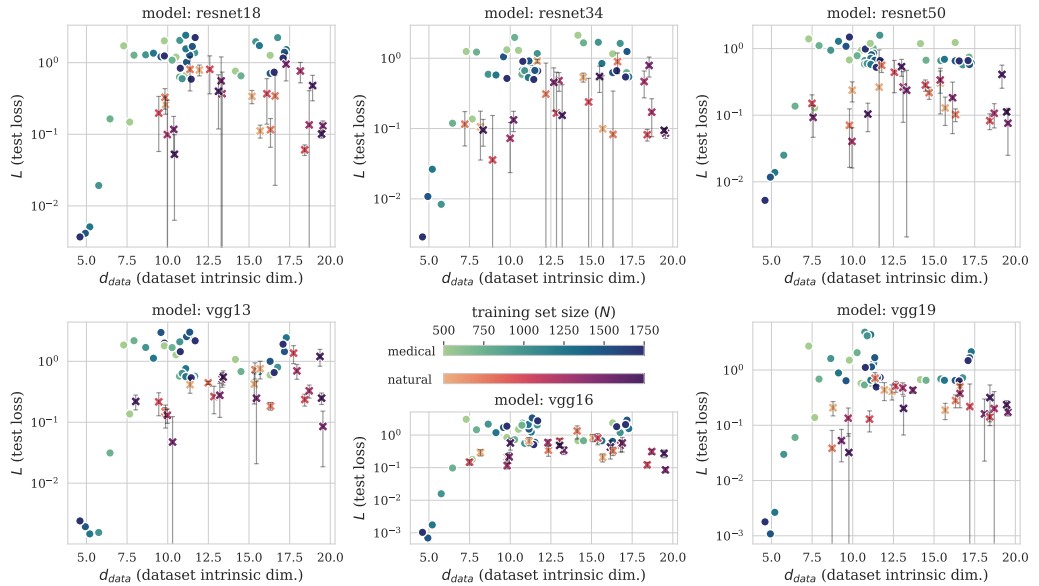

Figure 2: Scaling of log test set loss/generalization ability with training dataset intrinsic dimension ($d_{\text{data}}$) for natural and medical datasets. Each point corresponds to a (model, dataset, training set size) triplet. Medical dataset results are shown in blue shades, and natural dataset results are shown in red; note the difference in generalization error scaling rate between the two imaging domains. Standard deviation error bars are shown for natural image datasets for 5 different class pairs.

some test point $x_0 \in \mathcal{M}_{d_{\text{data}}}$ with true label $y = \mathcal{F}(x_0)$, the general goal of an adversarial attack is to find some $\tilde{x}$ that appears similar to $x_0$ — *i.e.*, $||\tilde{x} - x_0||_\infty$ is small — that results in a different, seemingly erroneous network prediction for $\tilde{x}$. Formally, the *robustness radius* of the trained network $f$ at $x_0$ is defined by

$$R(f, x_0) := \inf_{\tilde{x}} \left\{ ||\tilde{x} - x_0||_\infty : f(\tilde{x}) \neq y \right\}, \qquad (3)$$

where $x_0 \in \mathcal{M}_{d_{\text{data}}}$ (Zhang et al., 2021). This describes the largest region around $x_0$ where $f$ is robust to adversarial attacks. We define the *expected robust radius* of $f$ as $\hat{R}(f) := \mathbb{E}_{x_0 \sim \mathcal{M}_{d_{\text{data}}}} R(f, x_0)$.

**Theorem 3** (Adversarial Robustness and Label Sharpness Scaling Law). *Let $f$ be $K_f$-Lipschitz on $\mathbb{R}^n$. For a sufficiently large training set, the lower bound for the expected robustness radius of $f$ scales as $\hat{R}(f) \simeq \Omega\left(1/K_{\mathcal{F}}\right)$.*

*Proof.* This follows from Prop. 1 of Tsuzuku et al. (2018) — see Appendix A.4 for all details. $\square$

While it is very difficult to estimate robustness radii of neural networks in practice (Katz et al., 2017), we can instead measure the average loss penalty of $f$ due to attack, $\mathbb{E}_{x_0 \sim \mathcal{D}_{\text{test}}}(L(\tilde{x}) - L(x_0))$, over a test set $\mathcal{D}_{\text{test}}$ of points sampled from $\mathcal{M}_{d_{\text{data}}}$, and see if it correlates negatively with $\hat{K}_{\mathcal{F}}$ (Eq. (1)) for different models and datasets. As the expected robustness radius decreases, so should the loss penalty become steeper. We use FGSM (Goodfellow et al., 2015) attacks with $L_\infty$ budgets of $\epsilon \in \{1/255, 2/255, 4/225, 8/255\}$ to obtain $\tilde{x}$.

In Fig. 3 we plot the test loss penalty with respect to $\hat{K}_{\mathcal{F}}$ for all models and training set sizes for $\epsilon = 2/255$, and show the Pearson correlation $r$ between these quantities for each model, for all $\epsilon$, in Table 1 (per-domain correlations are provided in Appendix E.3). (We provide the plots for the other $\epsilon$ values, as well as for the test *accuracy* penalty, in Appendix E.3). Here we average results over the different training set sizes $N$ due to the lack of dependence of Theorem 3 on $N$.

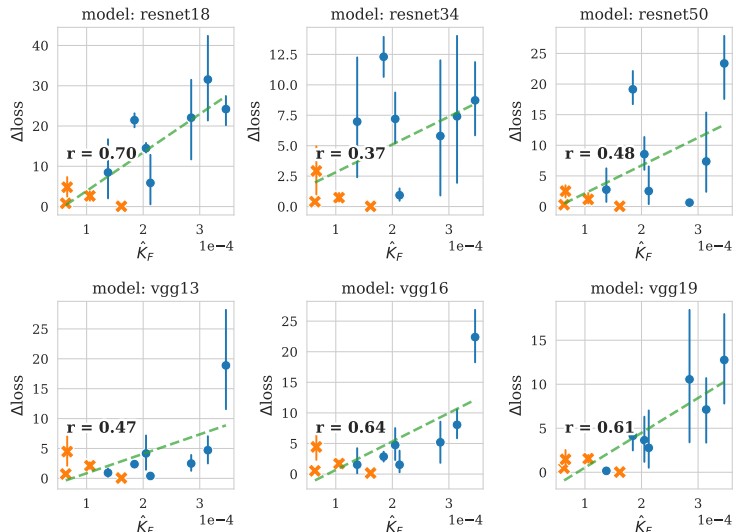

Figure 3: Test set loss penalty due to FGSM adversarial attack vs. measured dataset label sharpness ($\hat{K}_{\mathcal{F}}$) for models trained on natural and medical image datasets (orange and blue points, respectively). Pearson correlation coefficient $r$ also shown. Error bars are $95\%$ confidence intervals over all training set sizes $N$ for the same dataset.

As expected, the loss penalty is typically worse for models trained on datasets with higher $K_{\mathcal{F}}$, implying a smaller expected robustness radius. We see that medical datasets, which typically have higher $K_{\mathcal{F}}$ than natural datasets (Fig. 1), are indeed typically more susceptible to attack, as was found in Ma et al. (2021). In Appendix D.1 we show example clean

| Atk. $\epsilon$ | RN-18 | RN-34 | RN-50 | V-13 | V-16 | V-19 |
|---|---|---|---|---|---|---|
| 1/255 | 0.77 | 0.48 | 0.55 | 0.47 | 0.63 | 0.61 |
| 2/255 | 0.70 | 0.37 | 0.48 | 0.47 | 0.64 | 0.61 |
| 4/255 | 0.63 | 0.26 | 0.41 | 0.45 | 0.62 | 0.6 |
| 8/255 | 0.54 | 0.18 | 0.34 | 0.39 | 0.58 | 0.57 |

Table 1: Pearson correlation $r$ between test loss penalty due to FGSM attack and dataset label sharpness $\hat{K}_{\mathcal{F}}$, over all datasets and all training sizes. "RN" = ResNet, "V" = VGG.

and attacked images for each medical image dataset for $\epsilon = 2/255$. A clinical practitioner may not notice any difference between the clean and attacked images upon first look,[5] yet the attack makes model predictions completely unreliable. This indicates that adversarially-robust models may be needed for medical image analysis scenarios where potential attacks may be a concern.

## 7  CONNECTING REPRESENTATION INTRINSIC DIMENSION TO DATASET INTRINSIC DIMENSION AND GENERALIZATION

The scaling of network generalization ability with dataset intrinsic dimension $d_{\text{data}}$ (Sec. 5.1) motivates us to study the same behavior in the space of the network's learned hidden representations for the dataset. In particular, we follow (Ansuini et al., 2019; Gong et al., 2019) and assume that an encoder in a neural network maps input images to some $d_{\text{repr}}$-dimensional manifold of *representations* (for a given layer), with $d_{\text{repr}} \ll n$. As in the empirical work of Ansuini et al. (2019), we consider the intrinsic dimensionality of the representations of the final hidden layer of $f$. Recall that the test loss can be bounded above as $L = \mathcal{O}(K_L \max(K_f, K_{\mathcal{F}})N^{-1/d_{\text{data}}})$ (Thm. 1). A similar analysis can be used to derive a loss scaling law for $d_{\text{repr}}$, as follows.

**Theorem 4** (Generalization Error and Learned Representation Intrinsic Dimension Scaling Law). $L \simeq \mathcal{O}(K_L N^{-1/d_{\text{repr}}})$, *where $K_L$ is the Lipschitz constant for L.*

---

[5]That being said, the precise physical interpretation of intensity values in certain medical imaging modalities, such as Hounsfield units for CT, may reveal the attack upon close inspection.

We reserve the proof for Appendix A.3 due to length constraints, but the key is to split $f$ into a composition of an encoder and a final layer and analyze the test loss in terms of the encoder's outputted representations. Similarly to Eq. (2), $K_L$ is fixed for all experiments, such that we can simplify this result to $L \simeq \mathcal{O}(N^{-1/d_{\text{repr}}})$, *i.e.*,

$$\log L \lesssim -\frac{1}{d_{\text{repr}}} \log N + b \tag{4}$$

for some constant $b$. This equation is of the same form as the loss scaling law based on the *dataset* intrinsic dimension $d_{\text{data}}$ of Thm. 1. This helps provide theoretical justification for prior empirical results of $L$ increasing with $d_{\text{repr}}$ (Ansuini et al. (2019), as well as for it being similar in form to the scaling of $L$ with $d_{\text{data}}$ (Fig. (2)).

In Fig. 4 we evaluate the scaling of log test loss with the $d_{\text{repr}}$ of the training set (Eq. (4)), for each model, dataset, and training set size as in Sec. 5.1. The estimates of $d_{\text{repr}}$ are made using TwoNN on the final hidden layer representations computed from the training set for the given model, as in Ansuini et al. (2019). We also show the scaling of test accuracy in Appendix E.1, as well as results from using the MLE estimator to compute $d_{\text{repr}}$.

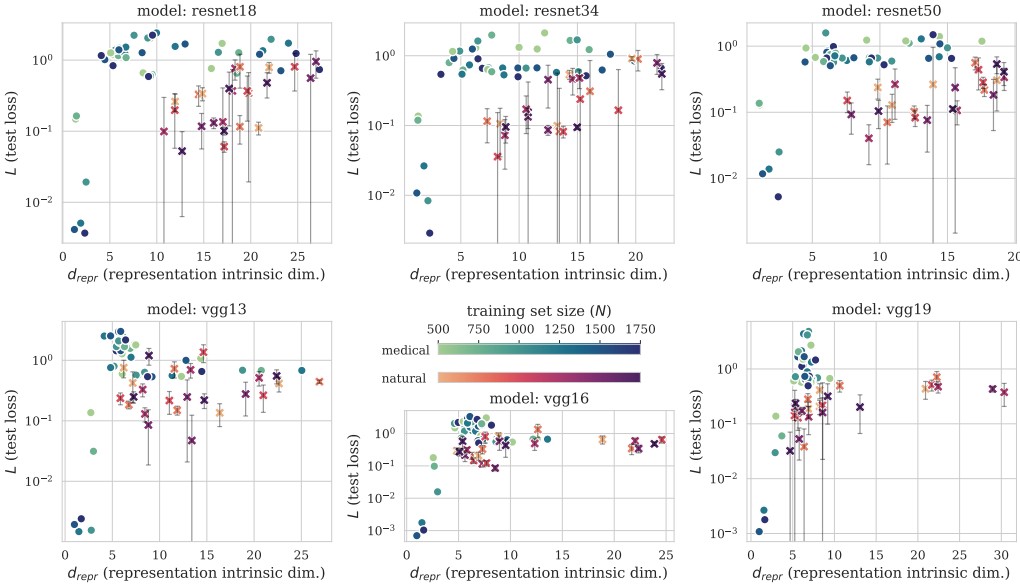

Figure 4: Scaling of log test set loss/generalization ability with the intrinsic dimension of final hidden layer learned representations of the training set ($d_{\text{repr}}$), for natural and medical datasets. Each point corresponds to a (model, dataset, training set size) triplet. Medical dataset results are shown in blue shades, and natural dataset results are shown in red.

We see that generalization error typically increases with $d_{\text{repr}}$, in a similar shape as the $d_{\text{data}}$ scaling (Fig. 2). The similarity of these curves may be explained by $d_{\text{repr}} \lesssim d_{\text{data}}$, or other potential factors unaccounted for. The former arises if the loss bounds of Theorems 1 and 4 are taken as *estimates*:

**Theorem 5** (Bounding of Representation Intrinsic Dim. with Dataset Intrinsic Dim.). *Let Theorems 1 and 4 be taken as estimates, i.e., $L \approx K_L \max(K_f, K_{\mathcal{F}})N^{-1/d_{\text{data}}}$ and $L \approx K_L N^{-1/d_{\text{repr}}}$. Then, $d_{\text{repr}} \lesssim d_{\text{data}}$.*

*Proof.* This centers on equating the two scaling laws and using a property of the Lipschitz constant of classification networks– see Appendix A.5 for the full proof. □

In other words, the intrinsic dimension of the training dataset serves as an upper bound for the intrinsic dimension of the final hidden layer's learned representations. While a rough estimate, we found this to usually be the case in practice, shown in Fig. 5 for all models, datasets and training

sizes. Here, $d_{\mathrm{repr}} = d_{\mathrm{data}}$ is shown as a dashed line, and we use the same estimator (MLE, Sec. 3.1) for $d_{\mathrm{data}}$ and $d_{\mathrm{repr}}$ for consistency (similar results using TwoNN are shown in Appendix E.2).

Intuitively, we would expect $d_{\mathrm{repr}}$ to be bounded by $d_{\mathrm{data}}$, as $d_{\mathrm{data}}$ encapsulates all raw dataset information, while learned representations prioritize task-related information and discard irrelevant details (Tishby & Zaslavsky, 2015), resulting in $d_{\mathrm{repr}} \lesssim d_{\mathrm{data}}$. Future work could investigate how this relationship varies for networks trained on different tasks, including supervised (*e.g.*, segmentation, detection) and self-supervised or unsupervised learning, where $d_{\mathrm{repr}}$ might approach $d_{\mathrm{data}}$.

## DISCUSSION AND CONCLUSIONS

In this paper, we explored how the generalization ability and adversarial robustness of a neural network relate to the intrinsic properties of its training set, such as intrinsic dimension ($d_{\mathrm{data}}$) and label sharpness ($K_{\mathcal{F}}$). We chose radiological and natural image domains as prominent examples, but our approach was quite general; indeed, in Appendix C.2 we evaluate our hypotheses on a skin lesion image dataset, a domain that shares similarities with both natural images and radiological images, and intriguingly find that properties of the dataset and models trained on it often lie in between these two domains. It would be interesting to study these relationships in still other imaging domains such as satellite imaging (Pritt & Chern, 2017), histopathology (Komura & Ishikawa, 2018), and others. Additionally, this analysis could be extended to other tasks (*e.g.*, multi-class classification or semantic segmentation), newer model architectures such as ConvNeXt (Liu et al., 2022), non-convolutional models such as MLPs or vision transformers (Dosovitskiy et al., 2021), or even natural language models.

Our findings may provide practical uses beyond merely a better theoretical understanding of these phenomena. For example, we provide a short example of using the network generalization dependence on label sharpness to rank the predicted learning difficulty of different tasks for the same dataset in Appendix C.3. Additionally, the minimum number of annotations needed for an unlabeled training set of images could be inferred given the measured $d_{\mathrm{data}}$ of the dataset and some desired test loss (Eq. (2)), which depends on the imaging domain of the dataset (Fig. 2).[6] This is especially relevant to medical images, where creating quality annotations can be expensive and time-consuming. Additionally, Sec. 6 demonstrates the importance of using adversarially robust models or training techniques for more vulnerable domains. Finally, the relation of learned representation intrinsic dimension to generalization ability (Sec. 7) and dataset intrinsic dimension (Theorem 5) could inform the minimum parameter count of network bottleneck layers.

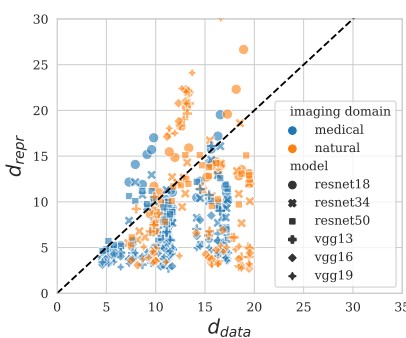

Figure 5: **Training set intrinsic dimension upper-bounds learned representation intrinsic dimension.** Each point corresponds to a (model, dataset, training set size) triplet.

A limitation of our study is that despite our best efforts, it is difficult to definitively say if training set label sharpness ($K_{\mathcal{F}}$) causes the observed generalization scaling discrepancy between natural and medical image models (Sec. 5.1, Fig. 2). We attempted to rule out alternatives via our formal analysis and by constraining many factors in our experiments (*e.g.*, model, loss, training and test set sizes, data sampling strategy, etc.). Additionally, we found that accounting for $K_{\mathcal{F}}$ in the generalization scaling law increases the likelihood of the law given our observed data (Appendix C.1). Altogether, our results tell us that $K_{\mathcal{F}}$ constitutes an important difference between natural and medical image datasets, but other potential factors unaccounted for should still be considered.

Our findings provide insights into how neural network behavior varies within and between the two crucial domains of natural and medical images, enhancing our understanding of the dependence of generalization ability, representation learning, and adversarial robustness on intrinsic measurable properties of the training set.

---

[6]Note that doing so in practice by fitting the scaling law model to existing ($L$, $N$, $d_{\mathrm{data}}$) results would require first evaluating a wider range of $N$ due to the logarithmic dependence of Eq. (2) on $N$.

AUTHOR CONTRIBUTIONS

N.K. wrote the paper, derived the mathematical results, ran the experiments, and created the visualizations. M.A.M. helped revise the paper, the presentation of the results, and the key takeaways.

ACKNOWLEDGMENTS

The authors would like to thank Hanxue Gu and Haoyu Dong for helpful discussion and inspiration.

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
