# Part I

# Supplementary Materials

## Table of Contents

## A  MATHEMATICAL DETAILS AND PROOFS

### A.1  EXTENSION OF RESULTS TO MULTI-CLASS CLASSIFICATION

**Generalization scaling laws.** Our results extend naturally from binary classification to multi-class classification. Given some test point $x_0$ of some unknown target class, if $x'_{tr}$ is the nearest neighbor to $x_0$ in the training set of the same class (both on $\mathcal{M}_{d_{\text{data}}}$), the term $\mathbb{E}_{\mathcal{D}_{\text{train}} \sim \mathcal{M}_{d_{\text{data}}}} ||x_0 - x'_{tr}||$ scales in expectation as

$$\mathcal{O}\left(\left(\frac{N+1}{C}\right)^{-1/d_{\text{data}}}\right) \simeq \mathcal{O}\left(\left(\frac{N}{C}\right)^{-1/d_{\text{data}}}\right) = \mathcal{O}(N^{-1/d_{\text{data}}}), \tag{5}$$

where $C$ is the total number of classes, assuming the classes to be evenly sampled in the training set. The same logic can be used for the intrinsic representation dimension $d_{\text{repr}}$ to show $\mathcal{O}\left(\left(\frac{N+1}{C}\right)^{-1/d_{\text{repr}}}\right) \simeq \mathcal{O}(N^{-1/d_{\text{repr}}})$. Therefore, the asymptotic upper bounds in the $d_{\text{data}}$ and $d_{\text{repr}}$ scaling laws (Theorems 1 and 4, respectively) still hold, as well as the derived result of Theorem 5.

**Label sharpness.** The label sharpness metric $\hat{K}_{\mathcal{F}}$ (Eq. 1) was formulated under the binary classification scenario, where data is either labeled with 0 or 1 (Sec. 3). However, it could potentially be extended to the multi-class scenario by simply replacing the $|y_j - y_k|$ term in the numerator of Eq. 1 with the indicator function $1_{y_j \neq y_k}$ as

$$\hat{K}_{\mathcal{F}} := \max_{j,k} \left( \frac{1_{y_j \neq y_k}}{||x_j - x_k||} \right), \tag{6}$$

which clearly simplifies to Eq. 1 for binary classification. This modification prevents $\hat{K}_{\mathcal{F}}$ from being biased by the numerical value of labels given to different classes, but a more careful extension could be pursued in the future to confirm a properly theoretically-motivated multi-class label sharpness metric.

## A.2 PROOF OF THEOREM 2 (APPROXIMATING $K_f$ WITH $K_{\mathcal{F}}$)

*Proof.* Let $x_1$ and $x_2$ be arbitrary datapoints sampled from $\mathcal{M}_{d_{\text{data}}}$, with nearest neighbors in the training set $\mathcal{D}_{\text{train}}$ of $\hat{x}_1$ and $\hat{x}_2$, respectively. Then,

$$
\begin{aligned}
|f(x_1) - f(x_2)| = |f(x_1) - f(x_2) + (\mathcal{F}(x_1) - \mathcal{F}(x_1) + \mathcal{F}(x_2) - \mathcal{F}(x_2) \\
+ f(\hat{x}_1) - f(\hat{x}_1) + f(\hat{x}_2) - f(\hat{x}_2))| \\
\leq |f(x_1) - f(\hat{x}_1)| + |f(x_2) - f(\hat{x}_2)| + |\mathcal{F}(x_1) - \mathcal{F}(x_2)| \\
+ |f(\hat{x}_1) - \mathcal{F}(x_1)| + |f(\hat{x}_2) - \mathcal{F}(x_2)|, \quad (7)
\end{aligned}
$$

by the triangle inequality. Because we assumed that $f(x) = \mathcal{F}(x) \, \forall x \in \mathcal{D}_{\text{train}}$, *i.e.*, the model is well-trained, the last two terms can be changed so that we have

$$
\begin{aligned}
|f(x_1) - f(x_2)| \leq |f(x_1) - f(\hat{x}_1)| + |f(x_2) - f(\hat{x}_2)| + |\mathcal{F}(x_1) - \mathcal{F}(x_2)| \\
+ |\mathcal{F}(\hat{x}_1) - \mathcal{F}(x_1)| + |\mathcal{F}(\hat{x}_2) - \mathcal{F}(x_2)|. \quad (8)
\end{aligned}
$$

Using the Lipschitz continuity of $f$ and $\mathcal{F}$, we have that

$$
\begin{aligned}
|f(x_1) - f(x_2)| \leq K_f(||x_1 - \hat{x}_1|| + ||x_2 - \hat{x}_2||) + K_{\mathcal{F}}(||x_1 - x_2|| + ||x_1 - \hat{x}_1|| + ||x_2 - \hat{x}_2||) \\
= K_{\mathcal{F}}||x_1 - x_2|| + (K_f + K_{\mathcal{F}})(||x_1 - \hat{x}_1|| + ||x_2 - \hat{x}_2||). \quad (9)
\end{aligned}
$$

Recall that the expected nearest-neighbor distance on $\mathcal{M}_{d_{\text{data}}}$ for some $N$ samples scales as $\mathcal{O}(N^{-1/d_{\text{data}}})$. Then, $\mathbb{E} ||x_1 - \hat{x}_1|| = \mathbb{E} ||x_2 - \hat{x}_2|| = \mathcal{O}((N + 1)^{-1/d_{\text{data}}}) \simeq \mathcal{O}(N^{-1/d_{\text{data}}})$. If we take the expectation of both sides of Eq. (9) over the training set, we can use this fact to obtain

$$\mathbb{E} \, |f(x_1) - f(x_2)| \leq K_{\mathcal{F}} \, \mathbb{E} \, ||x_1 - x_2|| + \mathcal{O}(\max(K_f, K_{\mathcal{F}})(N^{-1/d_{\text{data}}})). \tag{10}$$

But, the term on the right goes to zero as $N \rightarrow \infty$, so then $\Pr(|f(x_1) - f(x_2)| \leq K_{\mathcal{F}}||x_1 - x_2||) \rightarrow 1$ as $N \rightarrow \infty$, or in other words, the probability that $f$ is Lipschitz with the same constant $K_{\mathcal{F}}$ of $\mathcal{F}$. (A very similar proof can also be made to show that $\Pr(|\mathcal{F}(x_1) - \mathcal{F}(x_2)| \leq K_f||x_1 - x_2||) \rightarrow 1$ as $N \rightarrow \infty$). Therefore, the Lipschitz constant of $f$ converges to $K_{\mathcal{F}}$ in probability, or in other words, $K_f \rightarrow K_{\mathcal{F}}$. $\qquad\square$

## A.3 PROOF OF THEOREM 4 (GENERALIZATION ERROR AND REPRESENTATION INTRINSIC DIM. SCALING LAW)

*Proof.* Let $f$ be written as a composition of an encoder $g$, which outputs the final hidden representations of the input image, and a final output sigmoid (or softmax for multi-class classification) layer $h$, as $f = h \circ g$. Write the true label function $\mathcal{F}$ similarly, as some $\mathcal{F} = \mathcal{H} \circ \mathcal{G}$ for unknown $\mathcal{H}$ and $\mathcal{G}$ analogous to $h$ and $g$. Assume $h$ and $\mathcal{H}$ to be Lipschitz with respective constants $K_h$ and $K_{\mathcal{H}}$. Analogous to assuming $f(x) = \mathcal{F}(x)$ for all $x$ in the training set $\mathcal{D}_{\text{train}}$, posit a similar claim of $g(x) = \mathcal{G}(x) := z$, and $h(z) = \mathcal{H}(z), \forall x \in \mathcal{D}_{\text{train}}$.

Let $x$ be from the training set $\mathcal{D}_{\text{train}}$ with nearest neighbor (also in the training set) $\hat{x}$. Recall that we assume that $g(x) = \mathcal{G}(x) \, \forall x \in \mathcal{D}_{\text{train}}$, and that the loss vanishes at the true target label, as in Bahri et al. (2021). Let $z = g(x)$ and $\hat{z} = g(\hat{x})$.

Then, as we assumed $f$ and $\mathcal{F}$ to be Lipschitz,

$$\ell(f(x)) = |\ell(f(x)) - \ell(\mathcal{F}(x))| \leq K_L |f(x) - \mathcal{F}(x)| \tag{11}$$

$$= K_L |h(g(x)) - \mathcal{H}(\mathcal{G}(x))| = K_L |h(z) - \mathcal{H}(z)| \tag{12}$$

where $\ell(f(x))$ is the loss evaluated at a single datapoint, and the first equality is due to the loss vanishing at the true target label ($\ell(\mathcal{F}(x)) = 0$), and being non-negative. Continuing,

$$\ell(f(x)) \leq K_L |h(z) - \mathcal{H}(z)| \tag{13}$$

$$= K_L |h(z) - \mathcal{H}(z) + (h(\hat{z}) - h(\hat{z}) + \mathcal{H}(\hat{z}) - \mathcal{H}(\hat{z}))| \tag{14}$$

$$\leq K_L \left( |h(z) - h(\hat{z})| + |\mathcal{H}(z) - \mathcal{H}(\hat{z})| + |h(\hat{z}) - \mathcal{H}(\hat{z})| \right), \tag{15}$$

with the last line from the triangle inequality. As $h(z) = \mathcal{H}(z)$ for all $\{z = g(x) : x \in \mathcal{D}_{\text{train}}\}$, the last term vanishes, allowing us to write

$$\ell(f(x)) \leq K_L \left( |h(z) - h(\hat{z})| + |\mathcal{H}(z) - \mathcal{H}(\hat{z})| \right) \tag{16}$$

$$\leq K_L \left( K_h ||z - \hat{z}|| + K_\mathcal{H} ||z - \hat{z}|| \right) = K_L (K_h + K_\mathcal{H}) ||z - \hat{z}||, \tag{17}$$

so then

$$\ell(f(x)) \leq K_L \left( K_h ||z - \hat{z}|| + K_\mathcal{H} ||z - \hat{z}|| \right) = K_L (K_h + K_\mathcal{H}) ||z - \hat{z}||, \tag{18}$$

and

$$L = \underset{x \sim \mathcal{D}_{\text{test}}}{\mathbb{E}} \ell(f(x)) \leq K_L (K_h + K_\mathcal{H}) \underset{z, \hat{z} \sim \mathcal{D}_{\text{train}}}{\mathbb{E}} ||z - \hat{z}||, \tag{19}$$

where the rightmost expectation is taken over all $\{z = g(x) : x \in \mathcal{D}_{\text{train}}\}$ with corresponding nearest neighbor $\hat{z}$ (on the representation manifold). As the expectation of the nearest-neighbor distance of the representations on the manifold scales as $\mathcal{O}(N^{-1/d_{\text{repr}}})$, it follows that

$$L = \mathcal{O}(K_L \max(K_h, K_\mathcal{H}) N^{-1/d_{\text{repr}}}). \tag{20}$$

Because $h = \mathcal{H}$ on the training set representations, the same procedure as the proof for Theorem 2 can be used to show that $K_\mathcal{H} \simeq K_h$. Finally, note that the output layer $h$ was assumed to be a sigmoid. As the standard sigmoid (or softmax) layer is $1-$Lipschitz (Gao & Pavel, 2017), $K_\mathcal{H} \simeq K_h = 1$, so then

$$L \simeq \mathcal{O}(K_L N^{-1/d_{\text{repr}}}). \tag{21}$$

$\square$

### A.4 Proof of Theorem 3 (Adversarial Robustness and Label Sharpness Scaling Law)

*Proof.* Proposition 1 of (Tsuzuku et al., 2018) states that $\hat{R}(f, x_0) \geq M_{f,x_0}/(\sqrt{2}K_f)$ where $M_{f,x_0} > 0$ is the prediction margin, the difference between the target class prediction and the highest non-target class prediction of $f(x_0)$. Applying Thm. 2 given sufficiently large $N$ then gives $\hat{R}(f) = \mathbb{E}_{x_0 \sim \mathcal{M}_{d_{\text{data}}}} \hat{R}(f, x_0) \geq \mathbb{E}_{x_0 \sim \mathcal{M}_{d_{\text{data}}}} M_{f,x_0}/(\sqrt{2}K_f) = \Omega\left(1/K_f\right) \simeq \Omega\left(1/K_\mathcal{F}\right).$ $\square$

### A.5 Proof of Theorem 5 (Bounding of Representation Intrinsic Dim. with Dataset Intrinsic Dim.)

*Proof.* The estimation assumption implies that

$$K_L \max(K_f, K_\mathcal{F}) N^{-1/d_{\text{data}}} \approx K_L N^{-1/d_{\text{repr}}} \quad \Rightarrow \quad N^{-1/d_{\text{data}}} \approx \frac{N^{-1/d_{\text{repr}}}}{\max(K_f, K_\mathcal{F})}, \tag{22}$$

after which taking the logarithm of both sides gives

$$\begin{cases} d_{\text{repr}} \lesssim d_{\text{data}} & \text{if } K_f, K_\mathcal{F} \leq 1 \\ d_{\text{repr}} \gtrsim d_{\text{data}} & \text{otherwise,} \end{cases} \tag{23}$$

*i.e.*, $d_{\text{repr}} \lesssim d_{\text{data}}$ if the trained model $f$ and target model $\mathcal{F}$ are 1-Lipschitz (with respect to nearest neighbors in the training set).

Now, note that in our classification task setting, the decision boundaries/predictions of some $K$-Lipschitz network $f$ are the same as the $1-$Lipschitz version $\frac{1}{K}f$ (Béthune et al., 2022). As such, the scaling behavior we analyze here of $L$ vs. $d_{\mathrm{repr}}$ is the same as if $K_f = 1$. As $K_{\mathcal{F}} \simeq K_f$ (Theorem 2), Eq. (23) can be simplified to just $d_{\mathrm{repr}} \lesssim d_{\mathrm{data}}$. In practice, we also found that all datasets had $K_{\mathcal{F}} \ll 1$ (Fig. 1), so the first case of Eq. (23) should hold true anyways.

$\square$

# B ANALYSIS OF INTRINSIC DATASET PROPERTY CHARACTERISTICS (INTRINSIC DIMENSION AND LABEL SHARPNESS)

## B.1 INVARIANCE OF INTRINSIC DATASET PROPERTIES TO TRANSFORMATIONS

In Fig. 6, left, we show that measured dataset intrinsic dimension $d_{\mathrm{data}}$ estimates are barely affected by image resizing over a range of resolutions (square image sizes of $[32, 64, 128, 256, 512]$), with the specific example of $32 \times 32$ shown in the right of Fig. 7. We show the similar result of measured dataset label sharpness $\hat{K}_{\mathcal{F}}$ being invariant to image resizing in Fig. 6, right, and Fig. 8, right, besides all datasets' $\hat{K}_{\mathcal{F}}$ values being multiplied by the same positive constant (*i.e.*, the relative placement of the $\hat{K}_{\mathcal{F}}$ of each dataset stays the same with respect to such transformations). Because this constant is the same for all datasets for the given image resolution, it has no effect on the scaling law result of Eq. (2), as it can be folded into the constant $a$.

We show similar results for modifying the channel count of images (*i.e.*, modifying all grayscale images to RGB) in the left of Figs. 7 and 8.

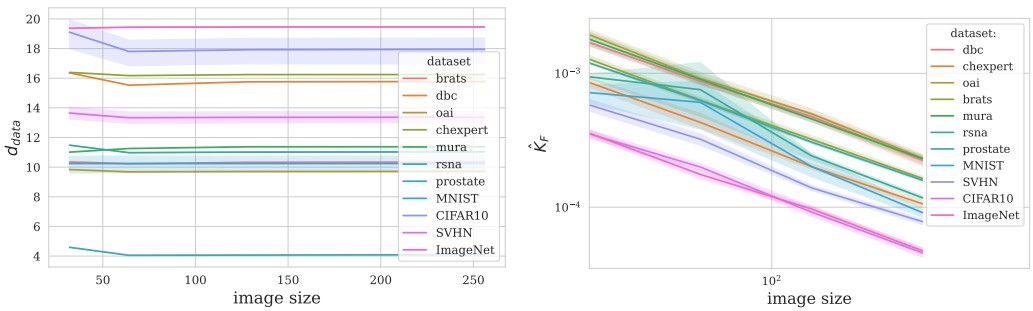

Figure 6: **Left:** Dependence of measured intrinsic dimension ($d_{\mathrm{data}}$) of the image datasets which we analyze with respect to image size (height and width). $d_{\mathrm{data}}$ values are averaged over all training set sizes; error bars indicate $95\%$ confidence intervals. **Right:** Same, but for measured dataset label sharpness $\hat{K}_{\mathcal{F}}$. $\hat{K}_{\mathcal{F}}$ values are averaged over all class pairings (Sec. 3.2); error bars indicate $95\%$ confidence intervals.

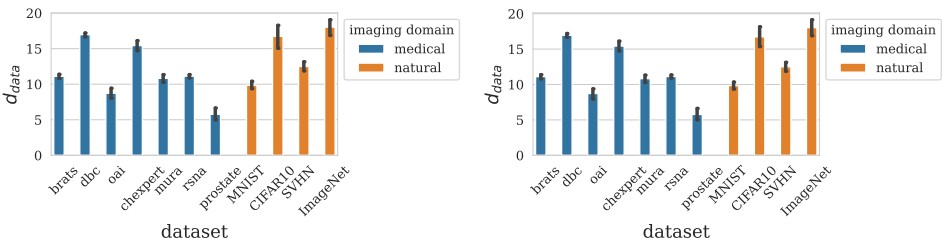

Figure 7: Measured intrinsic dimension ($d_{\mathrm{data}}$) of the natural (orange) and medical (blue) image datasets which we analyze (Sec. 4), for all images changed to RGB/3-channel (**left**), and all images resized to $32 \times 32$ (**right**). $d_{\mathrm{data}}$ values are averaged over all training set sizes; error bars indicate $95\%$ confidence intervals. Compare to the default results in Fig. 1, left ($224 \times 224$, original image channel counts) for reference.

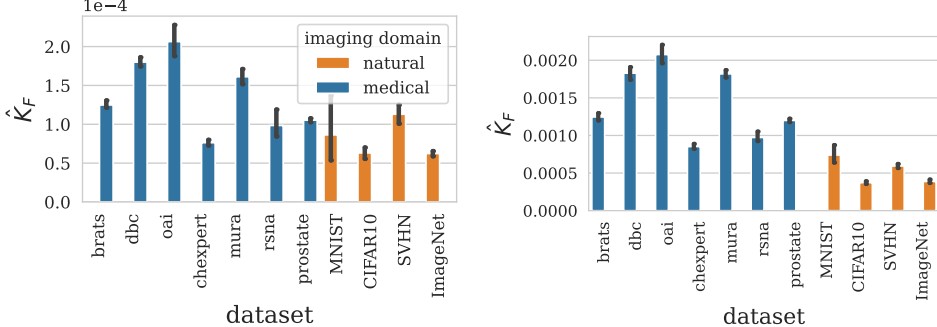

Figure 8: Measured label sharpnesses ($\hat{K}_{\mathcal{F}}$) of the natural (orange) and medical (blue) image datasets which we analyze (Sec. 4), for all images changed to RGB/3-channel (**left**), and all images resized to $32 \times 32$ (**right**). $\hat{K}_{\mathcal{F}}$ values are averaged over all class pairings (Sec. 3.2); error bars indicate 95% confidence intervals. Compare to the default results in Fig. 1, right ($224 \times 224$, original image channel counts) for reference.

## C  ADDITIONAL RESULTS, EXTENSIONS, AND APPLICATIONS

### C.1  LIKELIHOOD ANALYSIS OF THEORETICAL AND EMPIRICAL GENERALIZATION SCALING LAWS

We hypothesized in the main text that the observed discrepancies in generalization scaling between natural and medical images with respect to intrinsic dataset dimension $d_{\text{data}}$ (Fig. 2) were at least partially caused by the notable differences in dataset label sharpness ($K_{\mathcal{F}}$) between these two domains, indicated by our derived generalization scaling law of Equation (2). If we take Eq. (2) as an equality (in other words, a model that can be regressed to the observed generalization data in Fig. 2), we can analyze the likelihood that the observed shift between domains is caused by the scaling law's accounting for $K_{\mathcal{F}}$ by seeing if the likelihood of our scaling law model (Model **A**) which accounts for $K_{\mathcal{F}}$,

$$y_{\mathbf{A}}(d_{\text{data}}, N, K_{\mathcal{F}}; a) := \log L \simeq -\frac{1}{d_{\text{data}}} \log N + \log K_{\mathcal{F}} + a \tag{24}$$

is higher than the likelihood of a model that does not account for $K_{\mathcal{F}}$ (Model **B**),

$$y_{\mathbf{B}}(d_{\text{data}}, N; a) := \log L \simeq -\frac{1}{d_{\text{data}}} \log N + b. \tag{25}$$

Here, recall that $L$ is the test loss of a trained network given the intrinsic dimension $d_{\text{data}}$ and label sharpness $K_{\mathcal{F}}$ of the network's training dataset (Sec. 3), and $N$ is the size of the training set. Each of the two scaling law models **A** and **B** will be fit to the observed generalization scaling data $D$: $D = \{(L; d_{\text{data}}, N, K_{\mathcal{F}})_i\}_{\forall i}$ for model **A** and $D = \{(L; d_{\text{data}}, N)_i\}_{\forall i}$ for model **B**, using all result data $i$ for a given network architecture (*i.e.*, the datapoints in Fig. 2); the fitted parameters are $a$ and $b$, for each respective model. We obtained these fitted models using SciPy's `curve_fit` function (Virtanen et al., 2020), resulting in best-fit parameters of $\hat{a}$ and $\hat{b}$.

The likelihood ratio between two models is a well-known statistical test for determining the model that better explains the observed data (Vuong, 1989), and is defined by $\mathcal{R} := p(D|\text{model }\mathbf{A})/p(D|\text{model }\mathbf{B})$. For such regression problems, the likelihood ratio is evaluated as

$$\mathcal{R} = \frac{p(D|\text{model }\mathbf{A})}{p(D|\text{model }\mathbf{B})} = \frac{\exp\left[-\frac{1}{2}\sum_i (\log L_i - y_{\mathbf{A}}(d_{\text{data},i}, N_i, K_{\mathcal{F},i}; \hat{a}))^2\right]}{\exp\left[-\frac{1}{2}\sum_i (\log L_i - y_{\mathbf{B}}(d_{\text{data},i}, N_i; \hat{b}))^2\right]}. \tag{26}$$

Here, $\log \mathcal{R} > 0$ will indicate that model **A** explains the data better, $\log \mathcal{R} < 0$ will indicate that model **B** explains the data better, and $\log \mathcal{R} \approx 0$ indicates that neither model is preferred.

As shown in Table 2, we found that $\log \mathcal{R} > 0$ by a large margin for all network architectures, supporting the importance of accounting for $K_{\mathcal{F}}$ in the scaling law, due to the variability of it across

different domains. These results seem reasonable because as shown in Fig 2, there is a visible separation between the loss curves for the domains of natural and medical images. Allowing the scaling law to account for the label sharpness $K_\mathcal{F}$ of the dataset will make it more accurate because different datasets possess different $K_\mathcal{F}$ values (Fig. 1), and by Equation (24), different $K_\mathcal{F}$ values will move the loss curve up and down.

| ResNet-18 | ResNet-34 | ResNet-50 | VGG-13 | VGG-16 | VGG-19 |
|-----------|-----------|-----------|--------|--------|--------|
| 13.5 | 7.6 | 11.7 | 8.1 | 10.5 | 12.3 |

Table 2: Log-ratio $\log \mathcal{R}$ between (**A**) the likelihood of the network generalization $d_\text{data}$ scaling law model that accounts for label sharpness, and (**B**) the likelihood of the scaling law model that does not, given generalization data observed in our experiments (Fig. 2), for each network architecture.

### C.2 EVALUATING A DATASET FROM AN ADDITIONAL DOMAIN

In this section, we extend our analysis to a new dataset from a third domain beyond natural images and radiology images, in order to determine whether our hypotheses extend to other domains (*e.g.*, that dataset label sharpness is related to which domain the dataset is within). We use the ISIC skin lesion image dataset of Codella et al. (2018), which interestingly, has certain characteristics that both natural and radiological images share, such as being RGB photographs (like natural images), and having standardized acquisition procedure and object framing for the purpose of clinical tasks (like radiological images). For all experiments we use the task/labeling for melanocytic nevus detection.

First, we find that ISIC has an intrinsic dimension $d_\text{data} \simeq 12$ that is in between typical natural image dataset $d_\text{data}$ values and typical radiology dataset $d_\text{data}$ values (Fig. 9, left). We similarly see that its label sharpness $\hat{K}_\mathcal{F} \simeq 10^{-4}$ is in the upper end of typical natural image dataset $\hat{K}_\mathcal{F}$ values, and below all radiology dataset $\hat{K}_\mathcal{F}$ (Fig. 9, right). It makes intuitive sense that these intrinsic properties of the ISIC dataset are in between the two domains of natural and radiological images, given the aforementioned characteristics of images from both domains that it possesses.

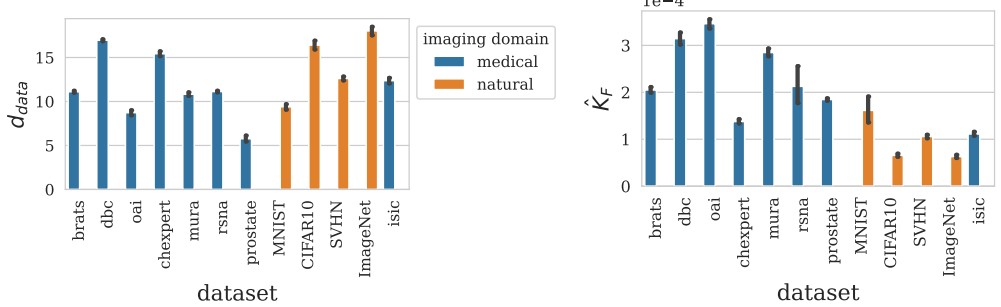

Figure 9: Measured intrinsic dimension ($d_\text{data}$, **left**) and label sharpnesses ($\hat{K}_\mathcal{F}$, **right**) of the natural (orange) and medical (blue) image datasets which we analyze (Sec. 4), **with the ISIC dataset included on the right of both figures.** $d_\text{data}$ values are averaged over all training set sizes, and $\hat{K}_\mathcal{F}$ over all class pairings (Sec. 3.2); error bars indicate $95\%$ confidence intervals.

We next performed the same generalization experiments as in the main text for ISIC, training each network model for the assigned task with $N = 1750$. Given our generalization scaling law of Eq. (2), ISIC having a $K_\mathcal{F}$ value between the typical respective values of natural and radiological domains would imply that models trained on the dataset would have test loss values between the models trained on these two domains, given ISIC's $d_\text{data}$. We see in Fig. 10 that this was indeed the case for all network architectures; the generalization ability of the ISIC models (indicated by purple circles) are between the typical generalization curves of natural image models and radiological image models.

Moreover, the "in-between" $K_\mathcal{F}$ of ISIC also implies that models trained on this dataset would be more adversarially robust than the radiological image models (with their high dataset $K_\mathcal{F}$ values),

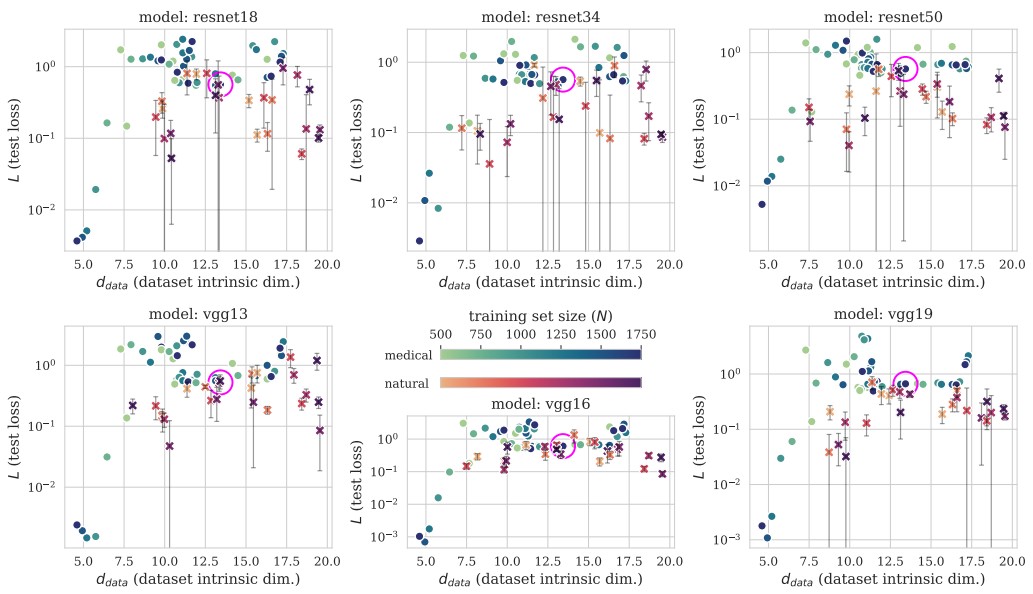

Figure 10: Same as Fig. 2, but with ISIC dataset results added with purple circles.

yet less robust than the natural image models (with their low dataset $K_{\mathcal{F}}$) (Theorem 3). In Fig. 11 we see that this is the case for some network architectures, while for others, ISIC models (purple circles) end up close to the natural image models.

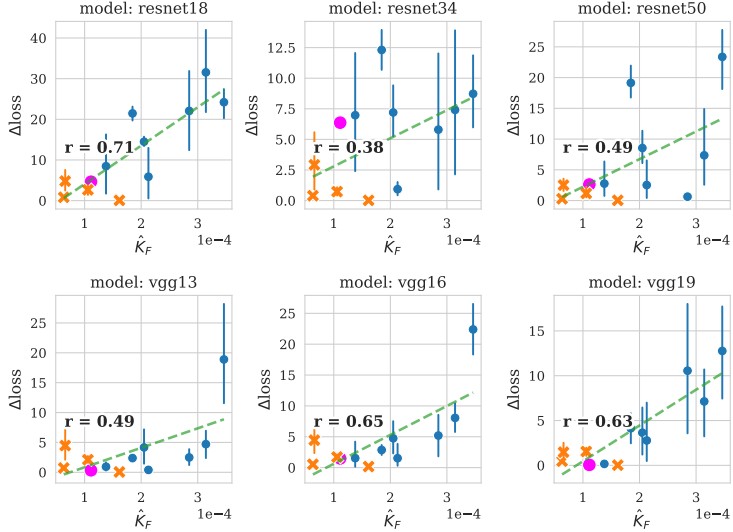

Figure 11: Same as Fig. 3, but with ISIC dataset results added with purple circles.

## C.3 PRACTICAL APPLICATION: TASK SELECTION FOR MEDICAL IMAGES

In this section we will demonstrate a practical usage of our formalism. It is common for new medical image datasets to come equipped with many different labels provided by clinical annotators, prior to any attempt to train a model to learn to make such predictions from the data. The question we examine in this section is: *given a new dataset with a variety of image labels, which tasks will be easier for a model learn, and which will be harder?* This is an important question to guide the model development process of practitioners who wish to take the first steps of training models for

automated diagnosis of a new dataset and/or modality, the answer of which may not be clear solely from the visible image characteristics.

For example, the RSNA-IH-CT dataset (Sec. 4) was annotated with labels for different types of hemorrhages, but some could be easier to detect than others. Consider that we wish to decide whether to train a binary classification model to (1) detect *any* type of hemorrhage out of 5 sub-types or (2) detect a specific type, such as epidural hemorrhage. Naïvely, it may seem that the second task is more specific and therefore may be more challenging, yet if some visual characteristic makes epidural hemorrhages easily noticeable, the first task could be more challenging, as it requires learning to differentiate between (a) healthy cases and (b) each type of hemorrhage. We can get a general idea for the relative difficulty of these two tasks using our derived scaling law, as follows.

Let's say that we wish to estimate which task is likely to be more challenging for a given model to learn by determining which has the higher expected test loss $L$. Our scaling law (Eq. (2)) estimates that $L \simeq \mathcal{O}(K_{\mathcal{F}} N^{-1/d_{\mathrm{data}}})$, but because the equation is a bound (not an equality), estimating absolute test loss values is not feasible. However, if we instead consider the *ratio* of test losses for two different possible tasks on the same dataset, a prediction is more tractable. While $N$ and $d_{\mathrm{data}}$ are both independent of task choice, the label sharpness $K_{\mathcal{F}}$ (Sec. 3.2) will change depending on the labels assigned to the data for the given task, which can be quickly measured from the dataset without any model training. If we take $K_{\mathcal{F}}^{(1)}$ and $L^{(1)}$ to be the measured label sharpness and expected test loss for the first task (detection of any hemorrhage), respectively, and likewise for $K_{\mathcal{F}}^{(2)}$ and $L^{(2)}$ for the second task (epidural hemorrhage detection), we get that approximately,

$$\frac{L^{(1)}}{L^{(2)}} \overset{\propto}{\sim} \frac{K_{\mathcal{F}}^{(1)} N^{-1/d_{\mathrm{data}}}}{K_{\mathcal{F}}^{(2)} N^{-1/d_{\mathrm{data}}}} = \frac{K_{\mathcal{F}}^{(1)}}{K_{\mathcal{F}}^{(2)}}, \tag{27}$$

implying that the task with the higher $K_{\mathcal{F}}$ will likely be more challenging for the model (higher test loss $L$).

To test this, we measured $\hat{K}_{\mathcal{F}}^{(1)} = 2.1 \pm 0.4 \times 10^{-4}$ and $\hat{K}_{\mathcal{F}}^{(2)} = 1.45 \pm 0.06 \times 10^{-4}$ for the two respective tasks (95% CI over 25 evaluations of $M^2$ pairings $M = 1000$, as in Sec. 3.2 and Fig. 1). Although approximate, Eq. (27) indicates that task 2 will be easier. We then trained each of our evaluated models for each of the two tasks, with results shown in Table 3 ($N = 1750$ and all other training details are the same as for the main paper experiments). We see that all models obtained lower test loss on task 2 than on task 1, and similarly obtained higher test accuracy, indicating that task 2 was indeed easier.

|  | ResNet-18 | ResNet-34 | ResNet-50 | VGG-13 | VGG-16 | VGG-19 | $\hat{K}_{\mathcal{F}}$ |
|---|---|---|---|---|---|---|---|
| **Task 1** | 1.29 | 1.23 | 1.03 | 0.69 | 0.50 | 0.51 | $2.1 \pm 0.4$ |
| **Task 2** | 0.64 | 0.66 | 0.66 | 0.63 | 0.62 | 0.90 | $1.45 \pm 0.06$ |
| **Task 1** | 76% | 74% | 74% | 73% | 75% | 76% | $2.1 \pm 0.4$ |
| **Task 2** | 80% | 83% | 83% | 85% | 82% | 81% | $1.45 \pm 0.06$ |

Table 3: **Top section:** Test set **loss** for each model trained on each of the two hemorrhage detection tasks, alongside the measured label sharpness $\hat{K}_{\mathcal{F}}$ for each task (Task 1 is detecting any hemorrhage, Task 2 is detecting epidural hemorrhage). **Bottom section:** Same, but for test set **accuracy**.

Note that Equation (27) is just an approximation, and that tasks with more similar measured $K_{\mathcal{F}}$ values for the same dataset could be harder to distinguish. Of course, this experiment is just an example, and future study with other datasets is warranted.

## C.4 EVALUATION AT MUCH HIGHER TRAINING SET SIZES

While many of our datasets do not support going to substantially higher training set sizes than our main experiments' maximum of $N = 1750$ (see Sec. 4), we can still evaluate the generalization scaling of models training on two datasets that do allow for significantly higher $N$. To this end, we trained each of our six models on the CheXpert medical image dataset and on the CIFAR-10 natural image dataset (for classes 1 and 2) at the highest training set size possible for binary classification on these datasets, $N = 9250$. We would expect from our generalization scaling law (Eq. (2)), that for a fixed dataset (and therefore $d_{\mathrm{data}}$ and $K_{\mathcal{F}}$) and architecture, the loss would decrease with higher

$N$. The results of this are shown in Tables 4 and 5 below; we see that this is indeed the case for all models (lower loss for higher training set size). We also see that the general trend of the natural image models having much lower loss than the medical image models is maintained, even though these two datasets have similar intrinsic dimensions ($d_{data} \simeq 15 - 17$).

| $N$ | ResNet-18 | ResNet-34 | ResNet-50 | VGG-13 | VGG-16 | VGG-19 |
|---|---|---|---|---|---|---|
| 9250 | 0.1660 | 0.1821 | 0.1179 | 0.1086 | 0.1045 | 0.0828 |
| 1000 | 0.5312 | 0.7402 | 0.5128 | 0.9764 | 0.6001 | 0.3974 |

Table 4: Test losses for models trained on CIFAR-10 binary classification for high training set size $N = 9250$ compared to those trained on $N = 1000$.

| $N$ | ResNet-18 | ResNet-34 | ResNet-50 | VGG-13 | VGG-16 | VGG-19 |
|---|---|---|---|---|---|---|
| 9250 | 0.7712 | 0.6370 | 0.6789 | 0.6014 | 0.6014 | 0.6016 |
| 1000 | 1.3479 | 0.7894 | 0.9793 | 0.6700 | 0.7409 | 0.6806 |

Table 5: Test losses for models trained on CheXpert binary classification for high training set size $N = 9250$ compared to those trained on $N = 1000$.

### C.5 DEPENDENCE OF NETWORK PERFORMANCE ON IMAGE RESOLUTION

It seems plausible that training a network to perform certain medical image binary classification tasks would be difficult at low image resolutions, due to the visual similarity of positive and negative images for some tasks (as any opposed to the typically low visual similarity of images from different classes in natural image datasets). To test this, we trained a ResNet-18 on each medical image dataset (with $N = 1750$ and all other training settings at their defaults) over a wide range of image resolutions (square image sizes of $[32, 64, 128, 256, 512]$), to see if the test accuracy was smaller for low resolutions. The results are shown in Fig. 12, and surprisingly, there is little performance drop for small resolutions. This may actually make sense, considering datasets like MedMNIST (Yang et al., 2023), where training for a wide variety of medical image classification tasks is possible even at $28 \times 28$ resolution. Of course, this would probably not be the case for more fine-grained tasks such as semantic segmentation.

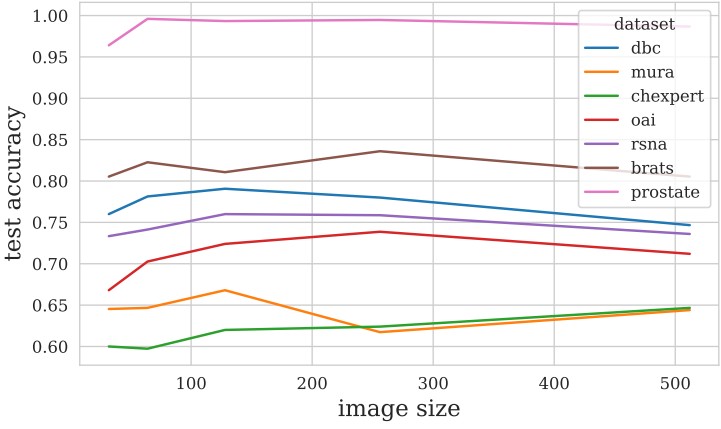

Figure 12: Dependence of network performance on image size for different medical image classification datasets (ResNet-18, training set size of 1750).

## D ADDITIONAL VISUALIZATIONS

### D.1 EXAMPLE ADVERSARIAL ATTACKS ON MEDICAL IMAGES

We show example attacked medical images for each dataset in Fig. 13.

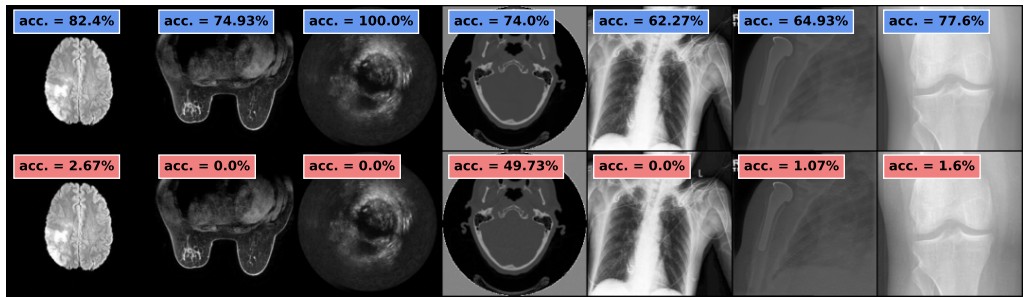

Figure 13: **Susceptibility of medical images to adversarial attack. Top row:** test set prediction accuracy of models trained on each medical image dataset for its corresponding diagnostic task (Sec. 4), with example test images shown. **Bottom Row:** accuracies after each test set was attacked by FGSM ($\epsilon = 2/255$), with example attacked images shown. The models are ResNet-18s with training set sizes of $N = 1750$.

# E    MAIN RESULTS WITH OTHER METRICS

In this section we will show our main results but with other metrics for generalization, adversarial robustness, and/or intrinsic dimensionality.

## E.1    GENERALIZATION SCALING WITH $d_{\text{data}}$ AND $d_{\text{repr}}$

**Continuation of Sec. 5.1.** In Fig. 14 we show the scaling of test *accuracy* with intrinsic dataset dimension $d_{\text{data}}$, using the default MLE estimator (Sec. 3.1). In Figs. 15 and 16 we show the scaling of test loss and accuracy, respectively, but instead using TwoNN (Sec. 3.1) to estimate $d_{\text{data}}$.

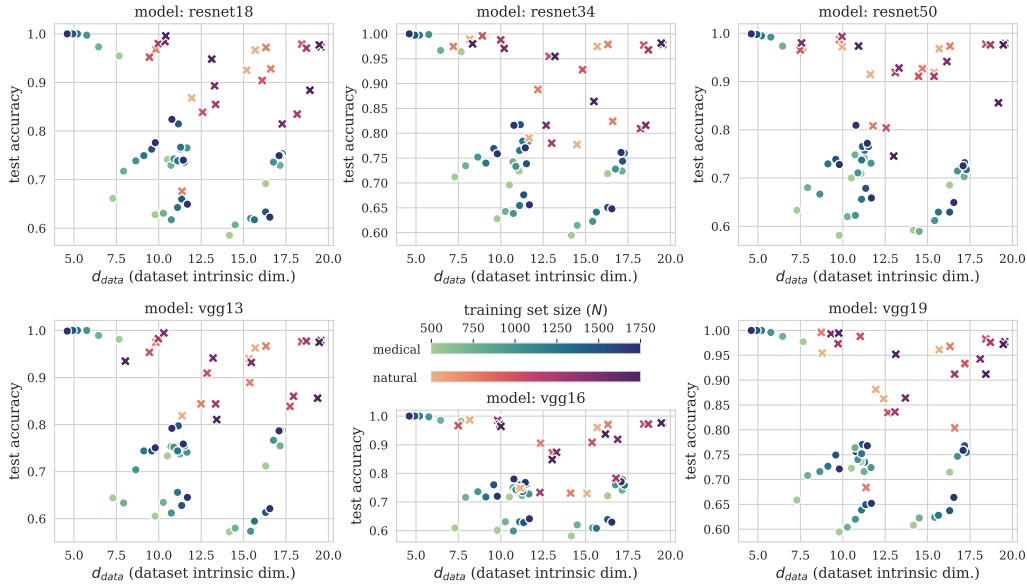

Figure 14: Scaling of test accuracy/generalization ability with training set intrinsic dimension ($d_{\text{data}}$) for natural and medical datasets.

**Continuation of Sec. 7.** Next, in Fig. 17 we show the scaling of test *accuracy* with learned representation intrinsic dimension $d_{\text{repr}}$, using the default TwoNN estimator (Sec. 3.1). In Figs. 18 and 19 we show the scaling of test loss and accuracy, respectively, but instead using MLE (Sec. 3.1) to estimate $d_{\text{repr}}$.

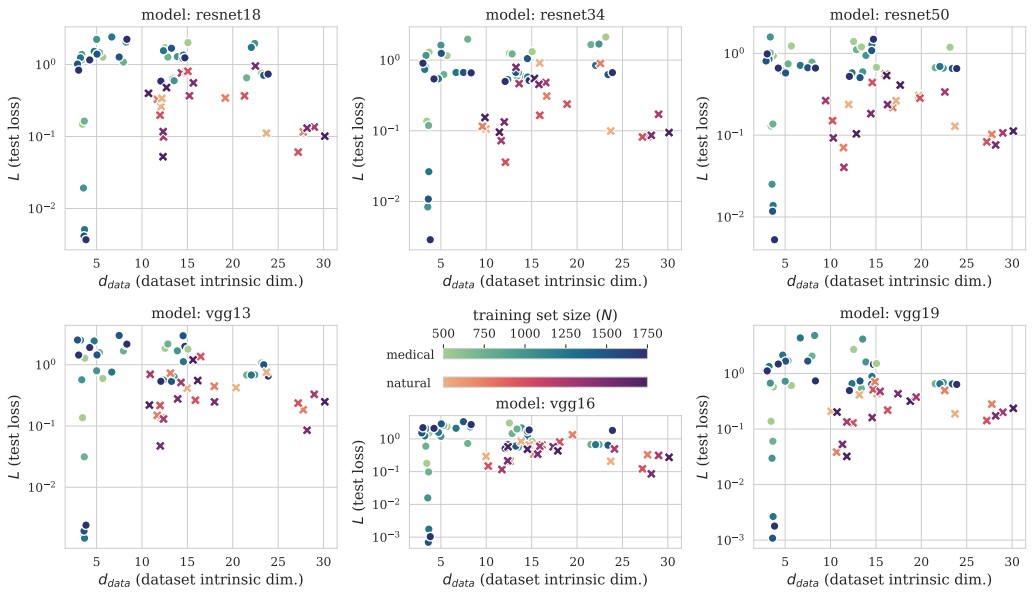

Figure 15: Scaling of log test loss/generalization ability with training set intrinsic dimension ($d_{\text{data}}$) for natural and medical datasets, with $d_{\text{data}}$ computed via TwoNN (Facco et al., 2017).

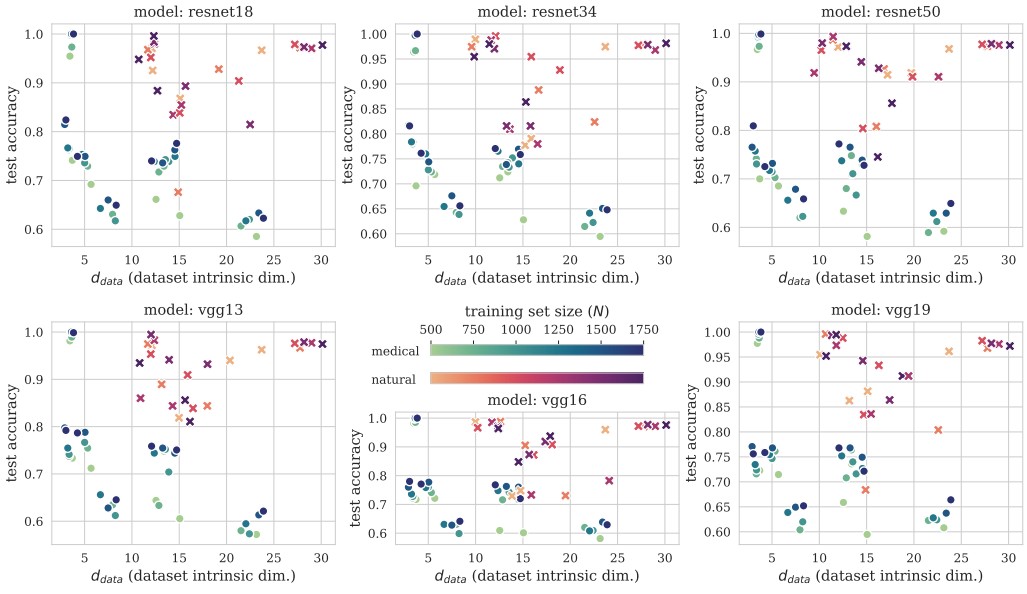

Figure 16: Scaling of test accuracy/generalization ability with training set intrinsic dimension ($d_{\text{data}}$) for natural and medical datasets, with $d_{\text{data}}$ computed via TwoNN (Facco et al., 2017).

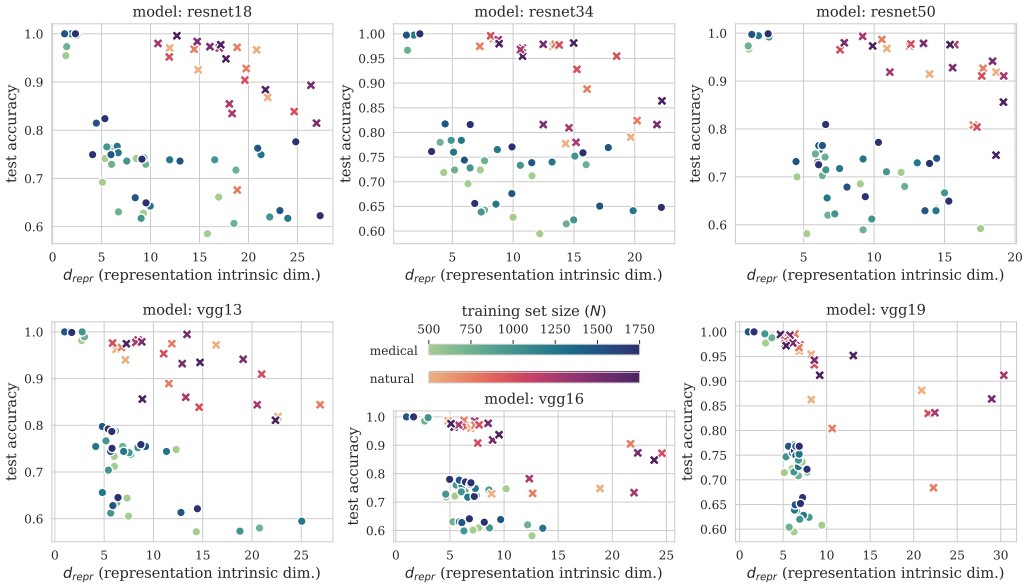

Figure 17: Scaling of test accuracy/generalization ability with the intrinsic dimension of final hidden layer learned representations of the training set ($d_{\mathrm{repr}}$) for natural and medical datasets.

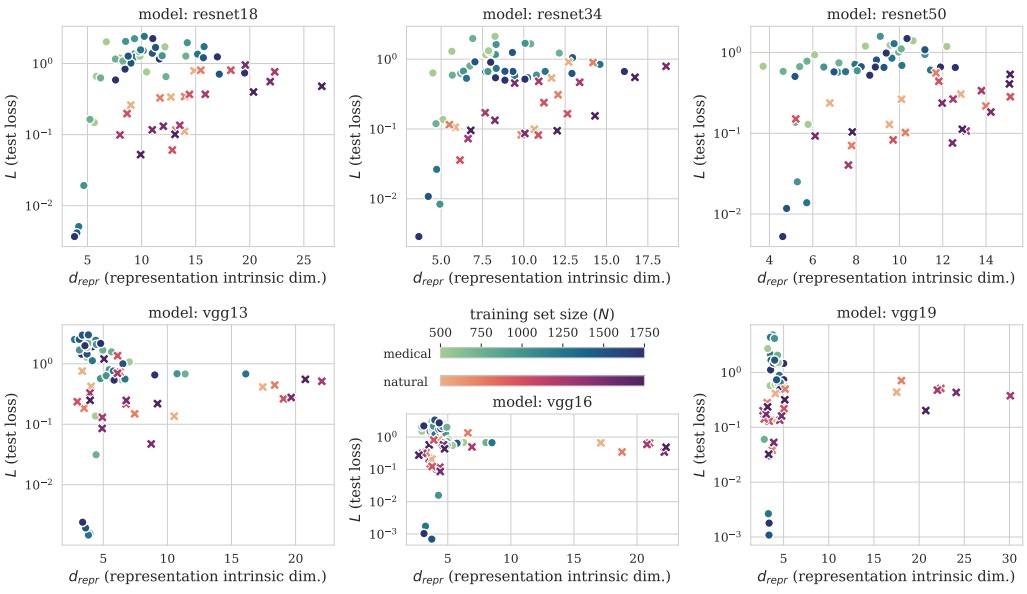

Figure 18: Scaling of log test loss/generalization ability with the intrinsic dimension of final hidden layer learned representations of the training set ($d_{\mathrm{repr}}$) for natural and medical datasets, with $d_{\mathrm{data}}$ computed via MLE (Sec. 3.1).

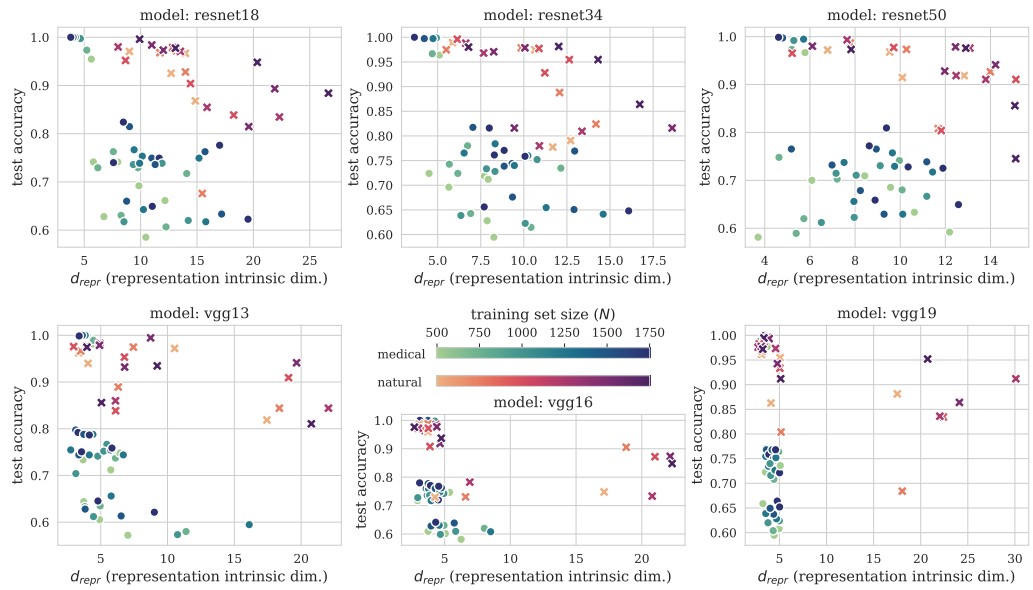

Figure 19: Scaling of test accuracy/generalization ability with the intrinsic dimension of final hidden layer learned representations of the training set ($d_{\mathrm{repr}}$) for natural and medical datasets, with $d_{\mathrm{data}}$ computed via MLE (Sec. 3.1).

## E.2 BOUNDING HIDDEN REPRESENTATION INTRINSIC DIMENSION WITH DATASET INTRINSIC DIMENSION

In Fig. 20 we show the $d_{\mathrm{data}}$ vs. $d_{\mathrm{repr}}$ results as in Fig. 5, but with dimensionality estimates computed with TwoNN instead of MLE (Sec. 3.1).

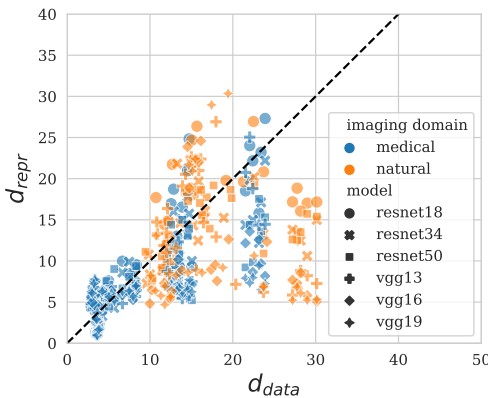

Figure 20: Training dataset intrinsic dimension $d_{\mathrm{data}}$ vs. learned representation intrinsic dimension $d_{\mathrm{repr}}$, both computed using TwoNN instead of MLE (Sec. 3.1). Each point corresponds to a (model, dataset, training set size) combination.

## E.3 ADVERSARIAL ROBUSTNESS SCALING WITH $\hat{K}_{\mathcal{F}}$

**Continuation of Sec. 6.** In Figs. 21, 22 and 23, we show the scaling of test loss penalty due to FGSM adversarial attack with respect to measured dataset label sharpness $\hat{K}_{\mathcal{F}}$, for attack $\epsilon$ of $1/255$, $4/255$, and $8/255$, respectively. In Figs 24, 25, 26 and 27 we instead show the scaling of test *accuracy* penalty, for each FGSM attack $\epsilon$ of $1/255$, $2/255$, $4/255$, and $8/255$, respectively.

Finally, in Tables 6 and 7 we report per-domain correlations of loss penalty and dataset $K_{\mathcal{F}}$, for medical images and natural images respectively.

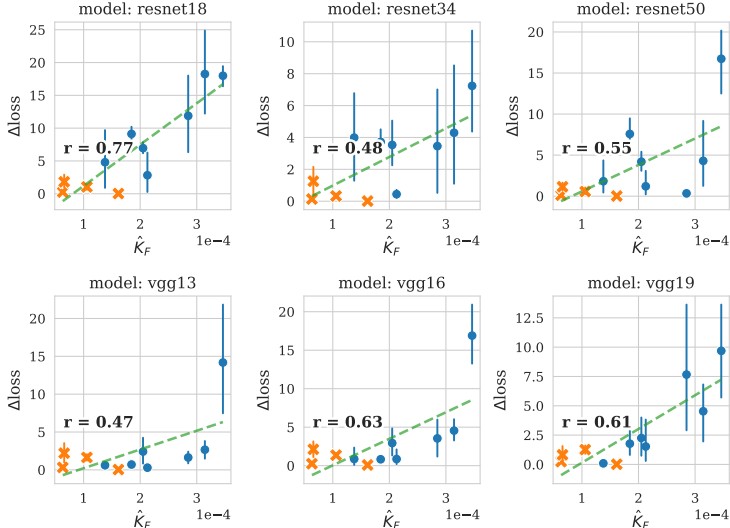

Figure 21: Scaling of test set loss penalty due to $\epsilon = 1/255$ FGSM adversarial attack with dataset label sharpness $K_{\mathcal{F}}$ for natural (orange) and medical (blue) datasets.

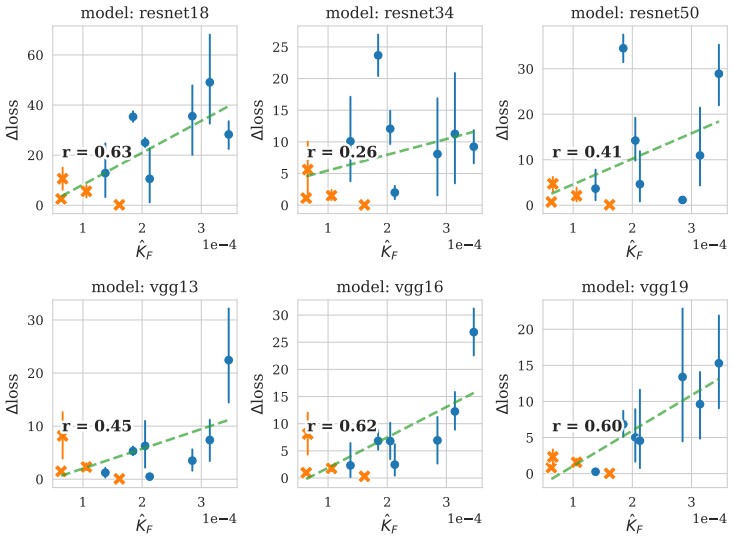

Figure 22: Scaling of test set loss penalty due to $\epsilon = 4/255$ FGSM adversarial attack with dataset label sharpness $K_{\mathcal{F}}$ for natural (orange) and medical (blue) datasets.

## F  TRAINING AND IMPLEMENTATIONAL DETAILS

This section provides training and implementation details beyond that of Sec. 4. We train all models with a binary cross-entropy loss function, optimize by Adam (Kingma & Ba, 2015) with a weight decay strength of $10^{-4}$ for 100 epochs. We use learning rates of $10^{-3}$ for ResNet models on all datasets, and $10^{-4}$ for VGG models on all datasets except SVHN, which required $10^{-6}$ to avoid loss divergence. ResNet-18, -34 and -50 models were trained with batch sizes of 200, 128, and 64, respectively, and 32 for all VGG models. We do not use any training image augmentations beyond

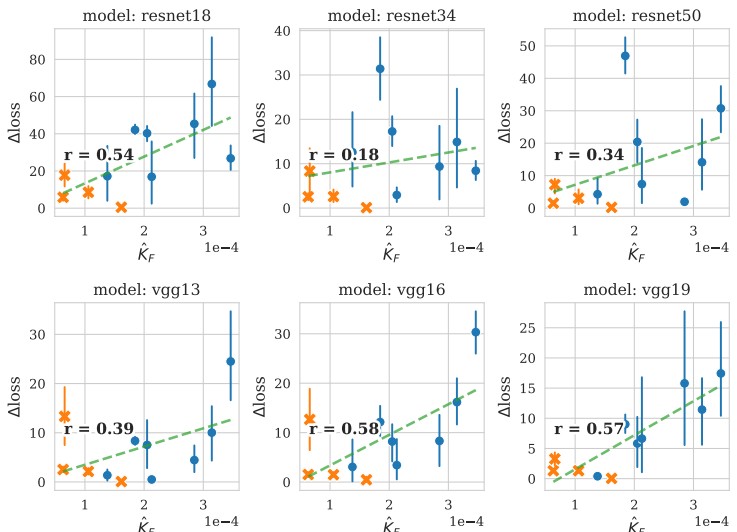

Figure 23: Scaling of test set loss penalty due to $\epsilon = 8/255$ FGSM adversarial attack with dataset label sharpness $K_{\mathcal{F}}$ for natural (orange) and medical (blue) datasets.

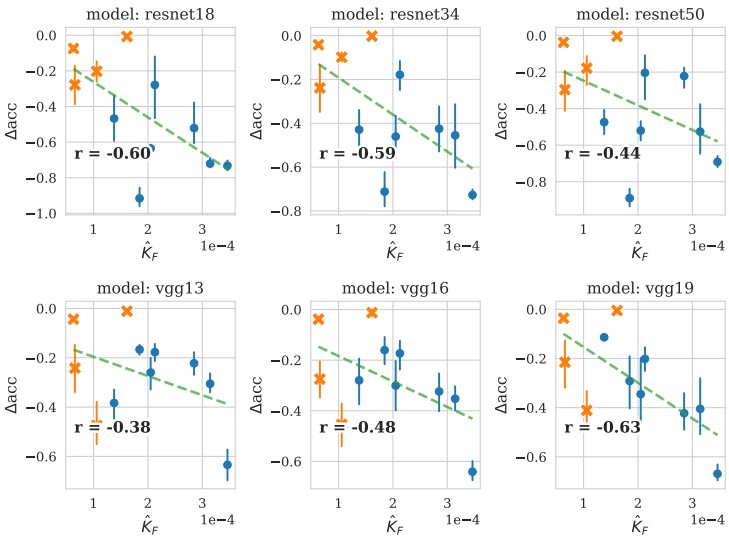

Figure 24: Scaling of test set accuracy penalty due to $\epsilon = 1/255$ FGSM adversarial attack with dataset label sharpness $K_{\mathcal{F}}$ for natural (orange) and medical (blue) datasets.

| Atk. $\epsilon$ | RN-18 | RN-34 | RN-50 | V-13 | V-16 | V-19 |
|---|---|---|---|---|---|---|
| $1/255$ | 0.67 | 0.26 | 0.43 | 0.55 | 0.69 | 0.6 |
| $2/255$ | 0.53 | 0.01 | 0.28 | 0.57 | 0.71 | 0.57 |
| $4/255$ | 0.41 | $-0.16$ | 0.14 | 0.56 | 0.7 | 0.53 |
| $8/255$ | 0.31 | $-0.23$ | 0.04 | 0.56 | 0.66 | 0.49 |

Table 6: Pearson correlation $r$ between test loss penalty due to FGSM attack and dataset label sharpness $\hat{K}_{\mathcal{F}}$, over all **medical image** datasets and all training sizes. "RN" = ResNet, "V" = VGG.

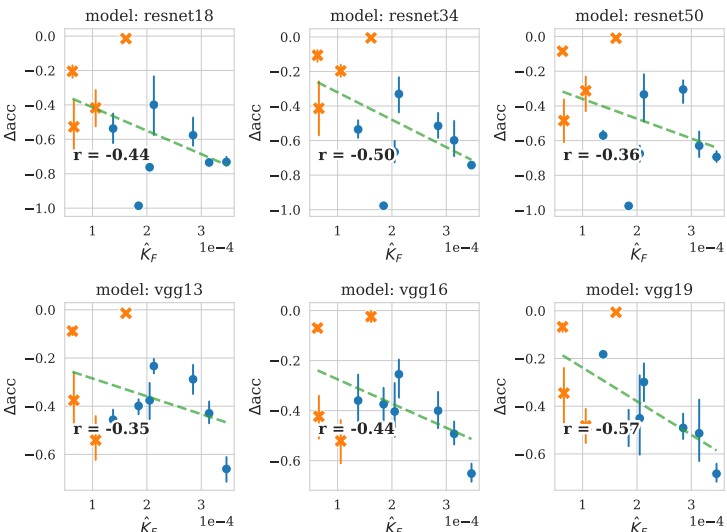

Figure 25: Scaling of test set accuracy penalty due to $\epsilon = 2/255$ FGSM adversarial attack with dataset label sharpness $K_{\mathcal{F}}$ for natural (orange) and medical (blue) datasets.

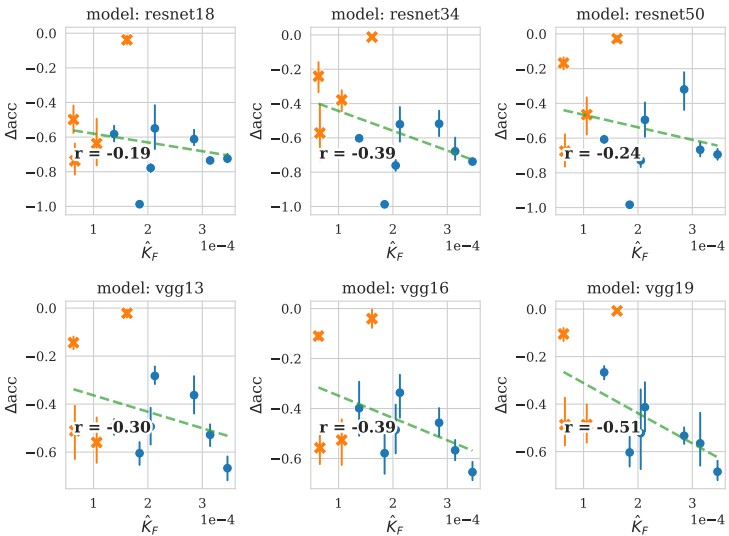

Figure 26: Scaling of test set accuracy penalty due to $\epsilon = 4/255$ FGSM adversarial attack with dataset label sharpness $K_{\mathcal{F}}$ for natural (orange) and medical (blue) datasets.

| Atk. $\epsilon$ | RN-18 | RN-34 | RN-50 | V-13 | V-16 | V-19 |
|---|---|---|---|---|---|---|
| 1/255 | $-0.39$ | $-0.37$ | $-0.39$ | $-0.36$ | $-0.38$ | $-0.24$ |
| 2/255 | $-0.42$ | $-0.37$ | $-0.41$ | $-0.42$ | $-0.41$ | $-0.36$ |
| 4/255 | $-0.49$ | $-0.41$ | $-0.44$ | $-0.47$ | $-0.43$ | $-0.53$ |
| 8/255 | $-0.58$ | $-0.47$ | $-0.48$ | $-0.5$ | $-0.43$ | $-0.66$ |

Table 7: Pearson correlation $r$ between test loss penalty due to FGSM attack and dataset label sharpness $\hat{K}_{\mathcal{F}}$, over all **natural image** datasets and all training sizes. "RN" = ResNet, "V" = VGG.

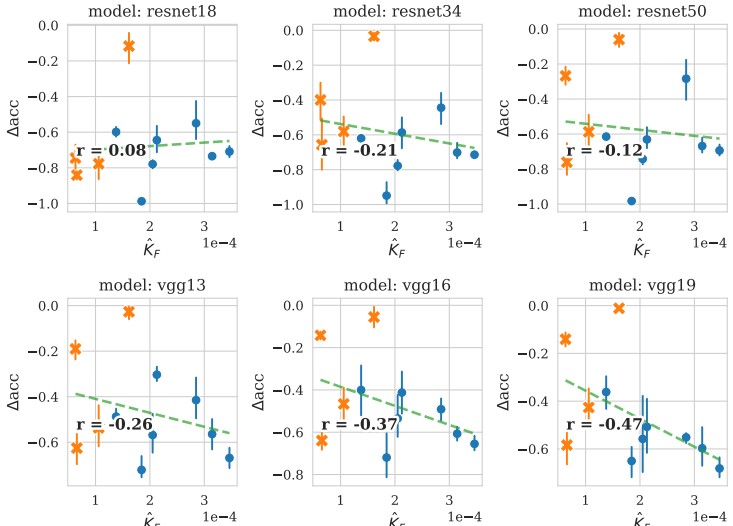

Figure 27: Scaling of test set accuracy penalty due to $\epsilon = 8/255$ FGSM adversarial attack with dataset label sharpness $K_{\mathcal{F}}$ for natural (orange) and medical (blue) datasets.

resizing to $224 \times 224$ and linear normalization to $[0, 1]$. We perform all experiments on a 48 GB NVIDIA A6000.

## G  MEDICAL IMAGE DATASET DETAILS

This section goes into full detail into the binary classification task definitions for each medical image dataset, beyond what is mentioned in Section 4. We follow the same task definitions for the medical image datasets as in Konz et al. (2022). Specifically:

- For OAI (Tiulpin et al., 2018), we use the screening packages 0.C.2 and 0.E.1, and define a negative class of X-ray images with Kellgren-Lawrence scores of 0 or 1, and a positive class of images with scores of 2+.

- For DBC (Saha et al., 2018), we use fat-saturated breast MRI slices. Slice images with a tumor bounding box label are positive, and any slice at least 5 slices away from a positive slice is negative.

- We use the same slice-labeling procedure as DBC for BraTS (Menze et al., 2014), for glioma labels in T2 FLAIR brain MRI slices.

- For Prostate MRI (Sonn et al., 2013), we use slices from the middle 50% of each MRI volume. Slices are labeled as negative if the volume's cancer risk score label is 0 or 1, and positive for 2+.

- For brain CT hemorrhage detection in RSNA-IH-CT (Flanders et al., 2020), we detect for *any* type of hemorrhage.