# OpenReview forum: "The Effect of Intrinsic Dataset Properties on Generalization: Unraveling Learning Differences Between Natural and Medical Images"
_ICLR.cc/2024/Conference — ICLR 2024 poster_

### Official Review · Reviewer_fAwr · 2023-10-30

**Soundness:** 4 excellent
**Presentation:** 3 good
**Contribution:** 2 fair
**Rating:** 6
**Confidence:** 4

**Summary:**

This paper tackles the problem of measuring dataset complexity for predicting test-set generalization, showing that the trends in natural images don't match the trends in medical images. This work offers three main contributions, with theoretical results provided throughout. The first is a proposed metric they refer to as "label sharpness", which is meant to serve as a proxy for difficulty in generalizing by looking at the worst-case scenario between two data points where the images are very similar (by the L2 norm) but the labels are different. Second, they demonstrate that the sharpness of the dataset label distribution is negatively correlated with the adversarial robustness of a neural network. The third contribution is connecting their notion of d_data to d_repr, and showing how their scaling laws are related, with d_data serving as an approximate upper bound. The authors do this over a variety of both medical and natural image datasets and architectures.

**Strengths:**

- The authors make an interesting attempt to provide a scaling law that is independent of the model itself, and rather depends on qualities of the data. Building insights which can be drawn from the training data alone seem very valuable.
- The authors investigate the effects of their proposed metrics thoroughly across a variety of architectures and datasets, at both training and inference.

**Weaknesses:**

- A central claim of this paper is that "medical images are typically more challenging to generalize to". I don't think that this is a widely held belief or has been consistently shown throughout the literature (to this the authors heavily cite a work from 2022, which shows that in some cases maybe the scaling laws are different).
- The measure of data sharpness, in the context of medical imaging, in my opinion is not well grounded. For medical images, especially those where you need to identify minor changes or tumors to correct classify things, the majority of the image will be the same as healthy images because the anatomies will be aligned.
- The results seem quite weak across models and datasets, and for some figures there are lines drawn in which don't represent proper trend lines (Figure 2 in particular). Additionally, in the adversarial setting, regression coefficients are computed across both medical/natural image datasets, even though it's clear that within the natural image datasets and medical image datasets that there is very little correlation (visually) (Figure 3, Figure 6).
- A claim was that models trained on datasets with similar data and of the same size N tend to perform much worse if the data is medical images, but the architectures chosen stem from the natural image domain and were tuned to those datasets.

**Questions:**

-  What was meant by "We ensure the standardization of KˆF across all datasets"? Is it not the case that each dataset has its own K^F?
- Why are there no natural image datasets without d_data less than 7.5? These seem like they would be important to draw trends with for scaling laws.
- I am very confused why there is a theorem saying that d_data is an upper bound for d_repr, when clearly the experimental results show that this isn't the case. What are the issues that cause this mismatch?
- Why is there not a clear trend with models that have trained on more data having lower test loss? (Figure 2)
- Taking the L2 norm between two high-dimensional images can result in near-zero distance even if the images are very different, why not work with distances of embeddings from pre-trained models? It would be helpful if there were examples of two images that are very close (by their metric) but have different labels.

---

> ### Author Response · Authors · 2023-11-18
> **Response to Reviewer fAwr (Part 1)**
>
> Thank you for your helpful suggestions, comments and questions. Responding to each in order and/or sorted by topic:
>
> > A central claim of this paper is that "medical images are typically more challenging to generalize to". I don't think that this is a widely held belief or has been consistently shown throughout the literature (to this the authors heavily cite a work from 2022, which shows that in some cases maybe the scaling laws are different).
>
> Thank you for your suggestion; accordingly we will remove this claim from the abstract to make it more accurate for camera-ready version if the paper is accepted (abstract modification isn't allowed for rebuttals).
>
> > The measure of data sharpness, in the context of medical imaging, in my opinion is not well grounded. For medical images, especially those where you need to identify minor changes or tumors to correct classify things, the majority of the image will be the same as healthy images because the anatomies will be aligned.
>
> > Taking the L2 norm between two high-dimensional images can result in near-zero distance even if the images are very different, why not work with distances of embeddings from pre-trained models? It would be helpful if there were examples of two images that are very close (by their metric) but have different labels.
>
> We considered computing $K_\mathcal{F}$ and/or $d_{data}$ in some feature space to lessen such effects, but beyond our desire to have $K_\mathcal{F}$ and $d_{data}$ being simple, model-independent metrics that can solely be computed from the raw dataset, there are also other reasons why we did not measure these in a feature space; please see our discussion on this in our response to Reviewer kQAF's related question.
>
> > The results seem quite weak across models and datasets, and for some figures there are lines drawn in which don't represent proper trend lines (Figure 2 in particular).
>
> We apologize for the confusion on this; these dotted lines were not meant to represent rigorous trend lines, but are used to show the general groupings of models trained in the two different domains, and to highlight that these groups are generally separate for the domains, motivating us to study these discrepancies. We have replaced them with test loss upper-bound regions for each domain which are created using the actual scaling law equation (Eq. 2), using the maximum measured dataset K_F for the given domain; please see the updated figure caption for more information.  Importantly, we now specify that they are still for illustrative purposes only.
>
> > Additionally, in the adversarial setting, regression coefficients are computed across both medical/natural image datasets, even though it's clear that within the natural image datasets and medical image datasets that there is very little correlation (visually) (Figure 3, Figure 6).
>
> Please see our response to Reviewer xVDy's similar question, where we address this and show our new results of evaluating the correlations within each domain.
>
> > A claim was that models trained on datasets with similar data and of the same size N tend to perform much worse if the data is medical images, but the architectures chosen stem from the natural image domain and were tuned to those datasets.
>
> We chose the ResNet and VGG architectures because they are also commonly used for medical image analysis; citations for this include f-AnoGAN (T. Schlegl et al., 2019) (an example seminal work), and review papers such as  "Deep convolutional neural network based medical image classification for disease diagnosis" (Yadav and Jadhav 2019),  and "Convolutional neural networks in medical image understanding: a survey" (Sarvamangala and Kulkarni, 2022), but many others could be found.
>
> > What was meant by "We ensure the standardization of KˆF across all datasets"? Is it not the case that each dataset has its own K^F?
>
> We apologize for the confusion; we meant that we minimize the effect of trivial dataset-specific factors on the $K_\mathcal{F}$ measurement by standardized image normalization, and by ensuring that $K_\mathcal{F}$ is functionally invariant to image resolution (Appendix E.1). We have updated the text to phrase this better.
>
> > Why are there no natural image datasets without d_data less than 7.5? These seem like they would be important to draw trends with for scaling laws.
>
> Good question; in our research, we were unable to find natural image (classification) datasets with very low values of $d_{data}$, that still represented realistic classification tasks, as that is the focus of our paper. Synthetic/toy datasets with controlled, low $d_{data}$ values would not have been within this focus.
>
> Please see our next response to your post for the responses to your remaining questions.

---

> > ### Comment · Reviewer_fAwr · 2023-11-20
> > **Response to Initial Rebuttal**
> >
> > Thank you for the helpful discussion, here are a few responses to your rebuttal to continue the conversation.
> >
> > 1.
> > >We considered computing and/or in some feature space to lessen such effects, but beyond our desire to have and being simple, model-independent metrics that can solely be computed from the raw dataset, there are also other reasons why we did not measure these in a feature space; please see our discussion on this in our response to Reviewer kQAF's related question.
> >
> > I understand that, given the current theorems, using the learned feature space might not be necessary, but as you point as in your response:
> >
> > > Although note that is the equivalent of doing this using the trained model's representations.
> >
> > It seems to me like the model that you choose to train with with play a large role in how well you generalize to the new domain (CNNs vs MLP vs Transformers), and that by making the metric model agnostic it makes any downstream scaling law more related to the abilities of CNNs than otherwise.
> >
> > 2.
> > >We apologize for the confusion on this; these dotted lines were not meant to represent rigorous trend lines, but are used to show the general groupings of models trained in the two different domains, and to highlight that these groups are generally separate for the domains, motivating us to study these discrepancies. We have replaced them with test loss upper-bound regions for each domain which are created using the actual scaling law equation (Eq. 2), using the maximum measured dataset K_F for the given domain; please see the updated figure caption for more information. Importantly, we now specify that they are still for illustrative purposes only.
> >
> > Thank you for clarifying this. I still think that the shaded regions, even though now labeled 'for illustrative purpose', are misleading and you should only include things in the figure that are functions of the data itself.
> >
> > 3.
> > >We chose the ResNet and VGG architectures because they are also commonly used for medical image analysis; citations for this include f-AnoGAN (T. Schlegl et al., 2019) (an example seminal work), and review papers such as "Deep convolutional neural network based medical image classification for disease diagnosis" (Yadav and Jadhav 2019), and "Convolutional neural networks in medical image understanding: a survey" (Sarvamangala and Kulkarni, 2022), but many others could be found.
> >
> > I understand that these models are also consistently used in medical image classification, but much like what reviewer MU2A stated, "the medical datasets used may not directly correspond to the natural image classification tasks", especially in terms of size of the datasets. Do these trends continue to hold as you get into significantly larger (where the majority of deep learning tasks live) data regimes?
> >
> > 4.
> > >As can be shown in Fig 6, most models have values greater than, supporting our results of upper-bounding (the area below the dashed line is where, which most models fall within). We apologize for not being clear enough with these results.
> >
> > Apologies but I am still confused about this. Even though a majority of the (model, dataset, training set size) triplets fall beneath the line, it seems that quite a few are above, especially those belonging to ResNet18 models which more than not seem to fall above the line. I think that these points to the fact it is not really an upper bound, but I could be missing something.
> >
> > For now I am maintaining my original score, but am willing to keep discussing the points I've raised above.

---

> ### Author Response · Authors · 2023-11-18
> **Response to Reviewer fAwr (Part 2)**
>
> > I am very confused why there is a theorem saying that d_data is an upper bound for d_repr, when clearly the experimental results show that this isn't the case. What are the issues that cause this mismatch?
>
> As can be shown in Fig 6, most models have $d_{data}$ values greater than $d_{repr}$, supporting our results of $d_{data}$ upper-bounding $d_{repr}$ (the area below the dashed line is where $d_{data} \geq d_{repr}$, which most models fall within). We apologize for not being clear enough with these results.
>
> > Why is there not a clear trend with models that have trained on more data having lower test loss? (Figure 2)
>
> Good question, we'd be happy to clarify. In Fig. 2, we do see that in some cases, models trained on more data (darker shaded points) do have lower test loss than the models with less data (lighter points). However, note that the loss dependence on training set size/$N$ is logarithmic (Equation 2), and because we have to limit ourselves to a relatively small range of training set sizes (see footnote 3 on page 3), the difference in loss due to training set size will be slight, and therefore may be overcome by other factors.

---

> ### Author Response · Authors · 2023-11-21
> **Response to Remaining Questions of Reviewer fAwr (Part 1)**
>
> Thank you for the quick reply, and for engaging in helpful further discussion! Responding to your comments and questions here:
>
> > It seems to me like the model that you choose to train with with play a large role in how well you generalize to the new domain (CNNs vs MLP vs Transformers), and that by making the metric model agnostic it makes any downstream scaling law more related to the abilities of CNNs than otherwise.
>
> Thank you for bringing up this point; we updated the introduction and discussion to be more clear that we focus on CNNs. This was partially because CNNs are more widely used than (the typically larger) vision transformers for medical image analysis in particular, due to the typically small size of medical image datasets, smaller total number of classes, etc, and MLPs don't usually see practical use for direct image analysis. While we don't have time to do so in the rebuttal, an interesting future work would be to see if the same scaling behavior is seen in non-convolutional models.
>
> > Thank you for clarifying this. I still think that the shaded regions, even though now labeled 'for illustrative purpose', are misleading and you should only include things in the figure that are functions of the data itself.
>
> Thank you for the suggestion;  we have accordingly simply removed the shaded regions. While we wanted this to be informative of the connection of our theoretical and empirical results, we certainly do not want to present anything that could be misleading.
>
> > I understand that these models are also consistently used in medical image classification, but much like what reviewer MU2A stated, "the medical datasets used may not directly correspond to the natural image classification tasks", especially in terms of size of the datasets. Do these trends continue to hold as you get into significantly larger (where the majority of deep learning tasks live) data regimes?
>
> Thank you for your question; to address it we have run new experiments to evaluate generalization scaling in the two domains at much higher training sizes. We trained each model on the CheXpert medical image dataset and on the CIFAR-10 natural image dataset (for classes 1 and 2) at the highest training set size possible for binary classification on these datasets, N=9250. We would expect from our scaling law (Eq. 2), that for a fixed dataset (and therefore d_{data} and $K_\mathcal{F}$) and architecture, the loss would decrease with higher $N$. The results of this are shown in the table below; we see that this is indeed the case for all evaluated models (lower loss for higher training set size).
>
> **Table 1: Test losses of CIFAR-10 models with different training set sizes $N$**:
> | $N$    | resnet18 | resnet34 | resnet50 | vgg13  | vgg16  | vgg19  |
> | ---- | -------- | -------- | -------- | ------ | ------ | ------ |
> | 9250 | 0.1660   | 0.1821   | 0.1179   | 0.1086 | 0.1045 | 0.0828 |
> | 1000 | 0.5312   | 0.7402   | 0.5128   | 0.9764 | 0.6001 | 0.3974 |
>
> **Table 2: Test losses of CheXpert models with different training set sizes $N$**:
> | $N$    | resnet18 | resnet34 | resnet50 | vgg13  | vgg16  | vgg19  |
> | ---- | -------- | -------- | -------- | ------ | ------ | ------ |
> | 9250 | 0.7712   | 0.6370   | 0.6789   | 0.6014 | 0.6014 | 0.6016 |
> | 1000 | 1.3479   | 0.7894   | 0.9793   | 0.6700 | 0.7409 | 0.6806 |
>
> We also see that the general trend of the natural image models having lower loss than the medical image models is maintained, even though these two datasets have similar intrinsic dimensions ($d_{data}\simeq 15$-$17$). As such, it appears that our findings extend to higher training set sizes, at least for these basic experiments. We added these results to a new Appendix B.7.
>
> We apologize for not having the time or resources to perform trainings of additional datasets or models (and as mentioned in the footnote of page 4, many of the datasets are too small to have such high training set sizes for binary classification), or extend to the other experiments, given the time constraints, which is why we here we just evaluated the primary generalization scaling result of Fig. 2/Equation 2/Section 5.

---

> ### Author Response · Authors · 2023-11-21
> **Response to Remaining Questions of Reviewer fAwr (Part 2)**
>
> > Apologies but I am still confused about this. Even though a majority of the (model, dataset, training set size) triplets fall beneath the line, it seems that quite a few are above, especially those belonging to ResNet18 models which more than not seem to fall above the line. I think that these points to the fact it is not really an upper bound, but I could be missing something.
>
> As we describe in Section 7, the result of $d_{repr} \lesssim d_{data}$ is a rough estimate, a theoretical result that came from making approximations. As such, it is not an exact upper bound (especially given the much higher extrinsic dimensionality of the raw data and learned representations), but a trend that we saw among the majority of the models. For example, we see that  $d_{repr} \leq d_{data}$ for 73% of all points, and  $d_{repr} \leq 1.25\times d_{data}$ for 83\%. As you point out, this seems less strong for the resnet18 models, which have 50\% and 67\% for these two respective metrics. This is why this result is not a focus or the main result of our text, and is placed near the end, as a potentially interesting, but simplified, extension of our main results. We included it in the paper because it points to a somewhat general trend between two easy-to-measure intuitive properties of the complicated distributions of raw data and learned representations, which seems to follow intuitions of how the complexity of learned networks representations should relate to the complexity of the raw data. A more exact bound could potentially be found from further analysis that includes other factors, but this is beyond the scope of our work.

---

> > ### Comment · Reviewer_fAwr · 2023-11-22
> > **Response to Rebuttal**
> >
> > Thank you for the discussion and clarifying points. In the context of the other reviews, and the very helpful discussion for clarifying my own understanding, I'm raising my score from a 3 to 6 and believe that the work and presentation is of conference-level quality

---

### Official Review · Reviewer_MU2A · 2023-11-01

**Soundness:** 3 good
**Presentation:** 3 good
**Contribution:** 3 good
**Rating:** 8
**Confidence:** 4

**Summary:**

The authors proposed a metric called label sharpness K_F to measure the similarity between images belonging to different classes in a dataset and investigated how this metric correlates with the generalisation performance and adversarial robustness. The new metric was validated on a variety of natural and medical image datasets through extensive experiments.

**Strengths:**

1. Present a theoretical understanding of the generalisation among natural and medical image datasets
2. Practical estimation of K_F
3. Validation through extensive experiments

**Weaknesses:**

1. Task dependence. The proposed label sharpness metric is inherently dependent on the specific tasks, whereas intrinsic dimension is a fixed data property. While the authors have made efforts to rule out alternatives, the medical datasets used may not directly correspond to the natural image classification tasks. For instance, the medical datasets tackle lesion detection where backgrounds are similar, likely making it more difficult than ImageNet classification. To better match ImageNet, the authors could also evaluate on RadImageNet by designing some easier tasks (e.g., CT/MRI/ultrasound discrimination). This also gives a benchmark on the scaling law when the training size is large.
2. Image preprocessing. All images were resized to 224x224 in this study, though prior work suggests intrinsic dimension is insensitive to the resolution of natural images. The authors should verify this holds for medical images, where a minimum resolution is needed to capture lesions. Providing comparisons across multiple resolutions could reveal if label sharpness or intrinsic dimension changes substantially.

**Questions:**

The proposed label sharpness metric K_F relies on having label information to calculate, yet the authors suggest it could guide estimates of required annotated data. It is unclear how K_F could be used for this purpose if labels are not yet available. The authors should clarify the practical utility of K_F for data annotation planning, since this appears unfeasible without existing labeled data. Potentially K_F could be estimated using a small initial labeled sample, but further analysis is needed on the variability of K_F to small labeled subsets.

---

> ### Author Response · Authors · 2023-11-18
> **Response to Reviewer MU2A**
>
> Thank you for your helpful review, feedback and questions. Responding to each in order:
>
> >Task dependence. The proposed label sharpness metric is inherently dependent on the specific tasks, whereas intrinsic dimension is a fixed data property. While the authors have made efforts to rule out alternatives, the medical datasets used may not directly correspond to the natural image classification tasks. For instance, the medical datasets tackle lesion detection where backgrounds are similar, likely making it more difficult than ImageNet classification. To better match ImageNet, the authors could also evaluate on RadImageNet by designing some easier tasks (e.g., CT/MRI/ultrasound discrimination). This also gives a benchmark on the scaling law when the training size is large.
>
> We appreciate these considerations and suggestions. While we did not have the time nor compute resources to perform implement and perform systematic training on and evaluation of a new dataset of that size, in relation to task dependence for medical images, we added experiments where we explore using our scaling law and label sharpness to estimate which tasks will be harder to learn prior to any model training for a new medical dataset, in Appendix F.
>
> > Image preprocessing. All images were resized to 224x224 in this study, though prior work suggests intrinsic dimension is insensitive to the resolution of natural images. The authors should verify this holds for medical images, where a minimum resolution is needed to capture lesions. Providing comparisons across multiple resolutions could reveal if label sharpness or intrinsic dimension changes substantially.
>
> Thank you for the suggestion; accordingly, we have now evaluated label sharpness and intrinsic dimension at a range of image resolutions, with the results shown in Fig. 20 of Appendix E.1. Notably, the intrinsic dimension is not affected, and K_F only changes by some constant that is the same for all datasets for a given resolution (which therefore has no effect on the generalization scaling behavior, as this constant can be folded into the a constant term in Equation 2).
>
> > The proposed label sharpness metric K_F relies on having label information to calculate, yet the authors suggest it could guide estimates of required annotated data. It is unclear how K_F could be used for this purpose if labels are not yet available. The authors should clarify the practical utility of K_F for data annotation planning, since this appears unfeasible without existing labeled data. Potentially K_F could be estimated using a small initial labeled sample, but further analysis is needed on the variability of K_F to small labeled subsets.
>
> Thank you for this practical consideration; please see our response to Reviewer xVDy's similar question about annotation count prediction.

---

### Official Review · Reviewer_kQAF · 2023-11-01

**Soundness:** 2 fair
**Presentation:** 3 good
**Contribution:** 3 good
**Rating:** 8
**Confidence:** 3

**Summary:**

The paper discusses the generalization discrepancies of deep neural networks between natural image datasets and medical image classification datasets. In particular, the authors derive a scaling law between the intrinsic dimension $d_\text{data}$ and its label sharpness $K_\mathcal{F}$ (describing how similar image pairs may be while still having different labels). They also derive a second scaling law between adversarial robustness of trained classifiers on a dataset and its label sharpness. The authors perform experiments using 4 natural image datasets and 7 medical imaging datasets and -to some extent- validate their laws empricially.

**Strengths:**

The paper is very well written and interesting to read. It tackles an interesting and important problem. Although not carefully checked, the Theorems and proofs seem sound.

**Weaknesses:**

1. **Image size** All datasets are resized to $HW=224\times224$. While this is a natural choice for natural image datasets, no practitioner would classify cancer or other abnormalities on such small images. However, the nearest-neighbor distance is dependent on $d_\text{data}$ and $d_\text{data}$ itself should also be dependent on $HW$. Couldn't this effect be the major explanation for the generalization error discrepancy we observe on real-world medical image data (also influencing $K_\mathcal{F}$)? In the Supplementary the authors only investigate the effect of much smaller resolutions ($32\times32$ px). I am not convinced this makes sense for medical imaging, and I doubt one could achieve above random accuracy on such small images ($d_\text{repr}$ could be dependent on this).

3. **Discrepancy between imaging domains** The analysis of the generalization discrepancies between imaging domains (Sec. 5.2) could be improved by computing the likelihood that the observed shift in Fig. 2 is indeed caused by Eq. 2 (similar for Fig. 5 and Eq. 4, respectively). In its current form it is hard to see that the derived law holds empirically. Particularly for Fig. 5, I doubt that this likelihood would be very high. Overall, I am not convinced that the conducted experiments support the theoretical results sufficiently. To some extent, the authors acknowledge this themselves, and openly discuss the fact that there are many (possibly much more contributing) other explanations for the observed discrepancy.

4. **Adversarial robustness** I find the analysis on adversarial robustness somewhat limited.
    - How does the scaling law behave w.r.t $K_\mathcal{F}$ estimates in the feature space of the trained classifier?
    - The chosen perturbation budgets $\epsilon \in \{ 1/255, 2/255, 4/255, 8/255\}$ have vastly different interpretations for medical and natural images because medical images are *always* represented using floating point numbers. For example for CT data a budget of $8/255$ on images rescaled to $[0,1]$ could alter the pixel values by several hundreds Houndsfield units (HU), a perturbation easily detectable due to the physical interpretation of HU.

5. **Investigate different architectures and tasks** Not that much of a weakness as it is a suggestion for future work. The authors limit their analysis to standard CNNs and a binary classification task. I think it would be very interesting to consider newer architectures and multi-class classification or even segmentation (also briefly mentioned by the authors w.r.t. relation between $d_\text{repr}$ and $d_\text{data}$) tasks. In particular it would be very interesting to see whether we can use these insights to predict uncertainty of a classification given the $K_\mathcal{F}$ between similar labels (e.g. types of metastasis).

6. **Source Code** Unfortunately the authors did not provide the source code for their experiments in the Supplementary.

**Questions:**

1. Is $\lVert x_j - x_k\rVert$ the $L_2$ distance measured in image space? Did you consider an extension where both $d_\text{data}$ and $K_\mathcal{F}$ are estimated in some feature representation (similar to what is done in Sec. 7) for $d_\text{repr}$?

2. How long would it take to compute $K_\mathcal{F}$ on larger images, e.g. $512\times 512$ which would be a common choice for the image matrix of CT images? Is it true that this only takes few seconds to compute $K_\mathcal{F}$ for large datasets such as ImageNet?

---

> ### Author Response · Authors · 2023-11-18
> **Response to Reviewer kQAF (Part 1)**
>
> Thank you for your helpful feedback. Responding to each in order:
>
> > **Image size** All datasets are resized to $HW=224\times 224$. While this is a natural choice for natural image datasets, no practitioner would classify cancer or other abnormalities on such small images. However, the nearest-neighbor distance is dependent on $d_{data}$ and $d_{data}$ itself should also be dependent on $HW$. Couldn't this effect be the major explanation for the generalization error discrepancy we observe on real-world medical image data (also influencing $K_\mathcal{F}$)? In the Supplementary the authors only investigate the effect of much smaller resolutions ($32\times 32$ px). I am not convinced this makes sense for medical imaging, and I doubt one could achieve above random accuracy on such small images ($d_{repr}$ could be dependent on this).
>
> Thank you for the helpful suggestions. To address these, we first performed experiments to determine how model performance depends on image resolution, for each medical image dataset, shown in Appendix B.6, over a range of resolutions between 512x512 and 32x32. We see that surprisingly, lowering resolution only slightly reduces test performance (if at all); we think that this is actually reasonable given that a wide variety of medical image classification tasks have been found to be feasible even at 28x28 (see the MedMNIST dataset, cited in Appendix B.6).
>
> In terms of the dependence of $d_{data}$ on image resolution itself, we evaluated $d_{data}$ for each dataset over a range of resolutions, and added the results to Fig. 24, left in Appendix E.1. We found that as expected, $d_{data}$ is invariant to image size.
>
> > **Discrepancy between imaging domains** The analysis of the generalization discrepancies between imaging domains (Sec. 5.2) could be improved by computing the likelihood that the observed shift in Fig. 2 is indeed caused by Eq. 2 (similar for Fig. 5 and Eq. 4, respectively). In its current form it is hard to see that the derived law holds empirically. Particularly for Fig. 5, I doubt that this likelihood would be very high. Overall, I am not convinced that the conducted experiments support the theoretical results sufficiently. To some extent, the authors acknowledge this themselves, and openly discuss the fact that there are many (possibly much more contributing) other explanations for the observed discrepancy.
>
> Thank you for your suggestion; accordingly, we have added a new section to the paper, Appendix B.4, where we analyze the likelihood that the observed shift between domains (shown in Fig 2) is caused by the scaling law's inclusion of $K_\mathcal{F}$ (Eq. 2), by showing that the likelihood of our scaling law model given the observed empirical results increases when $K_\mathcal{F}$ is accounted for, for all network architectures (this is possible if the scaling law is treated as an equality).
>
> For your second suggestion of Fig. 5 (now Fig. 4), we similarly fit the scaling law model (Eq. 4) to the data in Fig. 4, but as we mentioned that the reason for this scaling discrepancy between domains is unknown, this raw likelihood value is not really interpretable since there is not another model hypothesis likelihood to compare it to.
>
> Finally, to better illustrate the theoretical motivation for the empirical results, we have added possible test loss upper-bound illustrations for each domain to Figure 2, given the theoretical scaling law of Eq 2 (please see the updated caption as well as our response to Reviewer fAwr about this for more information).
>
> > **Adversarial robustness** I find the analysis on adversarial robustness somewhat limited.
> >> How does the scaling law behave w.r.t $K_\mathcal{F}$ estimates in the feature space of the trained classifier?
> >> The chosen perturbation budgets $\epsilon\in 1/255,2/255,4/255,8/255$ have vastly different interpretations for medical and natural images because medical images are always represented using floating point numbers. For example for CT data a budget of $8/255$ on images rescaled to $[0,1]$ could alter the pixel values by several hundreds Houndsfield units (HU), a perturbation easily detectable due to the physical interpretation of HU.
>
> For your first point, we found very little correlation between robustness and a $K_\mathcal{F}$ measured in the trained classifier's feature space (the same feature space as d_repr is computed). Please see our response to your question below ("Is $||x_j-x_k||$ the $L_2$ distance…") for more discussion about measuring $K_\mathcal{F}$ in feature space, where we discuss the motivations for measuring $K_\mathcal{F}$ in image space rather than a feature space.
>
> Thank you for bringing up this second point; we agree that the attack perturbations in the medical images may be visible upon close inspection by a trained practitioner. We have updated the last paragraph of Sec. 6 to address this (and softened our claim of how easily the attack could be noticed).
>
> Response continued in next comment.

---

> ### Author Response · Authors · 2023-11-18
> **Response to Reviewer kQAF (Part 2)**
>
> > **Investigate different architectures and tasks** Not that much of a weakness as it is a suggestion for future work. The authors limit their analysis to standard CNNs and a binary classification task. I think it would be very interesting to consider newer architectures and multi-class classification or even segmentation (also briefly mentioned by the authors w.r.t. relation between $d_{repr}$ and $d_{data}$) tasks. In particular it would be very interesting to see whether we can use these insights to predict uncertainty of a classification given the $K_\mathcal{F}$ between similar labels (e.g. types of metastasis).
>
> We appreciate the suggestions for future work and agree that they would be valuable; we amended the Discussion section (first paragraph) to discuss these as well.
>
> > **Source Code** Unfortunately the authors did not provide the source code for their experiments in the Supplementary.
>
> We apologize for not making our code availability clearer. We will make the code available upon acceptance.
>
> > Is $||x_j-x_k||$ the $L_2$ distance measured in image space? Did you consider an extension where both $d_{data}$ and $K_\mathcal{F}$ are estimated in some feature representation (similar to what is done in Sec. 7) for $d_{repr}$?
>
> Our apologies for not being more clear on this; $||x_j-x_k||$ is the $L_2$ distance measured in image space for $d_{data}$ and $K_\mathcal{F}$. We measure $d_{data}$ in image space rather than in a representation space of some encoder so that it is a model-independent metric of the dataset, although note that $d_{repr}$ is the equivalent of doing this using the trained model's representations. We have updated the text to be more clear on this. While $K_\mathcal{F}$ could be measured in some feature space, there is no $K_\mathcal{F}$-like value that appears in the generalization scaling law with respect to the learned feature space (Theorem 4 / Equation 4), so there is no current theoretical motivation for it, as opposed to the standard $K_\mathcal{F}$ metric which appears in the image-space generalization scaling law (Equation 2). Unlike our $K_\mathcal{F}$ measured in image space, such a $K_\mathcal{F}$ value measured in feature space would also be challenging to standardize due to it being sensitive to the distribution of activation values for the given network layer, which could vary between architectures, or models of the same architecture trained on different datasets.
>
> > How long would it take to compute $K_\mathcal{F}$ on larger images, e.g. $512\times512$ which would be a common choice for the image matrix of CT images? Is it true that this only takes few seconds to compute $K_\mathcal{F}$ for large datasets such as ImageNet?
>
> At 512x512 image resolution, with the default settings ($M$ = 1000, in Sec. 3.2) the $K_\mathcal{F}$ of a dataset takes 1-2 seconds to compute (fully parallelized) on our GPU for a given class pairing/binary classification task labeling.
>
> Thank you for allowing us to clarify with the second question. The compute time is unaffected by the size of the base dataset because $M$ class-balanced random samples from the base dataset are used to create $M^2$ estimates in Equation 1, rather than using the entire dataset, because as mentioned in Section 3.2, we found that $M$ values greater than 1000 hardly affected the value of $K_\mathcal{F}$. This is why the size of the base dataset that is being sampled from (e.g, ImageNet) does not have an effect.

---

> > ### Comment · Reviewer_kQAF · 2023-11-21
> >
> > I would like to thank the authors for providing such a detailed rebuttal and for conducting additional analysis on the likelihood ratio and the effect on image size. I would like to increase my score and vote for acceptance of the manuscript in its current form.

---

### Official Review · Reviewer_xVDy · 2023-11-05

**Soundness:** 2 fair
**Presentation:** 2 fair
**Contribution:** 3 good
**Rating:** 6
**Confidence:** 4

**Summary:**

The paper hypothesizes about and investigates the reason behind generalization challenges in medical imaging domain. The intrinsic dimension of data is known to affect the generalization error; however, the strength of this relationship seems to depend on the data being from medical or natural imaging domains. This may hint to a missing component in the known scaling laws; the authors propose and empirically validate a scaling law where the discrepancy between these domains can be partially explained by the difference in their “label sharpness”. Furthermore, the authors connect the label sharpness to adversarial robustness.

**Strengths:**

- The paper addresses an important knowledge gap in understanding generalization error of medical imaging domain.
- The proposed scaling law is investigated both theoretically and empirically on a wide range of models.
- The paper is clearly written and relatively easy to follow.

**Weaknesses:**

- The proposed scaling law and the effect of label sharpness is only (partially) verified in two domains. I suggest adding other domains or datasets that have different label sharpness values.
- The empirical results for the adversarial examples are not convincing enough, specially if we only consider the datasets within a single domain. Also the relationship between test set loss penalty vs label sharpness varies wildly based on the model used. This makes it harder for the reader to understand the message of section 6.
- The theoretical and empirical results may have limited practical uses. Adding some results on use cases such as determining the minimum number of annotations (as mentioned in Discussion) can further strengthen the paper.

**Questions:**

- How does Figure 1 look like for datasets within the medical domain? It seems that there are at least three datasets (brats, chexpert, prostate) with similar label sharpness to natural imaging domain (MNIST). Do you see some sort of similar separation that you see between natural and medical imaging domains?
- Are there any other domains that you can use to validate your hypothesis further?
- Can you provide the confidence intervals for at least one of the subplots in Figure 2?
- The results in Figure 3, considering only the results in each group, is not that convincing. Can you report the correlation within each group?
- There are numerous similar examples of adversarial attacks natural imaging domain to the ones presented in Figure 4. What is special about these examples compared to the ones presented in even the earliest papers in this field (e.g., Szegedy et al. 2013, Intriguing properties of neural networks)?
- Can you provide some intuition on why in Figure 5 and for the VGG models, the representation intrinsic dimension doesn’t change that much for medical imaging domain?

---

> ### Author Response · Authors · 2023-11-18
> **Response to Reviewer xVDy (Part 1)**
>
> Thank you for your helpful feedback, comments and questions. Responding to each in order and/or sorted by topic:
> > The proposed scaling law and the effect of label sharpness is only (partially) verified in two domains. I suggest adding other domains or datasets that have different label sharpness values.
>
> > Are there any other domains that you can use to validate your hypothesis further?
>
> To evaluate our hypothesis in an imaging domain beyond natural images or radiology images, we have performed new experiments using the ISIC skin lesion image dataset (Codella et al., ISBI 2018) for the task of melanocytic nevus detection, as it has unique characteristics that both natural and radiological imaging domains also share.
>
> From this we added the following results to the new Appendix B.5 and referenced them in the text. Notably, we see that this dataset has a $K_\mathcal{F}$ value of about 1.6E-4, between the typical values for (a) most radiology datasets and (b) most natural image datasets (Appendix Fig. 20, right). We similarly see that when Fig 1 is updated with the generalization vs. d_data results for models trained on this dataset (shown in Appendix Fig. 21), that ISIC models typically lie between the two typical generalization curves of medical and natural images; we also have similar findings for the adversarial attack results for some architectures (Appendix Fig. 22). We think that such properties of the ISIC dataset and models trained on it being "in between" the natural image and radiology image domains is reasonable, given that the dataset possesses characteristics from both imaging domains (such as the images being framed for standardized medical diagnosis, yet being RGB photographs).
>
> > The empirical results for the adversarial examples are not convincing enough, specially if we only consider the datasets within a single domain. Also the relationship between test set loss penalty vs label sharpness varies wildly based on the model used. This makes it harder for the reader to understand the message of section 6.
>
> > The results in Figure 3, considering only the results in each group, is not that convincing. Can you report the correlation within each group?
>
> We posit that the test loss penalty vs. label sharpness relationship varying between models could be partially due to the dependence of (FGSM) adversarial attack on the model itself (which is why we presented separate results for each model). Next, we have added the per-group correlation results as you suggested in Appendix B.3 (Tables 2 and 3). While the medical image datasets still typically exhibit a moderate positive correlation of loss penalty with $K_\mathcal{F}$, the natural image dataset correlations are typically close to zero or even somewhat positive. We suspect that this is because there are many other factors which contribute to adversarial attack success beyond $K_\mathcal{F}$, which appear to dominate over the relatively thin range of $K_\mathcal{F}$ values which the natural image datasets possess. Still, our results are meant as an approximate "rule of thumb", where given a model architecture, the adversarial susceptibility will typically increase with $K_\mathcal{F}$, which is a simple, model-free statistic to measure, and so can provide useful information given its ease of obtaining. Moreover, for all models, on average the medical image datasets are more prone to attack than the natural image datasets by a noticeable margin; presenting such learning discrepancies between imaging domains like this that are made visible using simple dataset properties like $K_\mathcal{F}$ is one of the focuses of our paper.
>
> Please proceed to our next comment for the next part of our response.

---

> ### Author Response · Authors · 2023-11-18
> **Response to Reviewer xVDy (Part 2)**
>
> > The theoretical and empirical results may have limited practical uses. Adding some results on use cases such as determining the minimum number of annotations (as mentioned in Discussion) can further strengthen the paper.
>
> Thank you for your helpful suggestion; in response we added Appendix F to demonstrate a practical use of our formalism that is common for medical imaging datasets: predicting which task will be harder to train a model for given a new dataset with different labels/possible tasks to choose from.
>
> In terms of the annotation count prediction suggestion, we ran experiments to predict the number of annotations needed ($N$) to achieve some desired loss on for a given network by measuring the $d_{data}$ of the unlabeled dataset; i.e., fitting the scaling law model of equation 2 to the existing (log loss, $d_{data}$, $N$, $K_\mathcal{F}$) data from our experiments, and then extrapolating the $N$ needed to achieve some log loss given the measured new dataset $d_{data}$, and using the average $K_\mathcal{F}$ of the given imaging domain. However, because of the logarithmic dependence on $N$, the scaling law hardly varies over the range of our observed data with respect to $N$, as we work over a relatively small range of training sizes (see footnote 3 on page 3), resulting in noisy extrapolations given the lack of model performance data over a larger range of $N$. This is also why our paper is not focused on the dependence on $N$/samples complexity, but instead intrinsic dataset properties. We have updated the discussion on practical applications to reflect this.
>
> > How does Figure 1 look like for datasets within the medical domain? It seems that there are at least three datasets (brats, chexpert, prostate) with similar label sharpness to natural imaging domain (MNIST). Do you see some sort of similar separation that you see between natural and medical imaging domains?
>
> Our apologies in advance if we misunderstood your question, but we suspect that the reason MNIST has a label sharpness higher than the other natural datasets, and similar to the lower end of certain medical datasets (brats, chexpert, prostate), is because MNIST shares certain similarities to the medical datasets, such as having one channel, and having fairly standardized object location and illumination, on a dark background.
>
> > Can you provide the confidence intervals for at least one of the subplots in Figure 2?
>
> As requested, we have added confidence intervals to both Fig 2 and Fig 4 for the natural image results, in the form of standard deviation error bars over 5 different binary classification class pairings for the given {dataset/model/training set size triplet} (as described in Section 4, we did not perform multiple training runs with different class pairs for the medical image datasets because they only have two classes, and so resulted in very little performance change with different runs). Please note that certain points have long-tailed error bars (also visibly exacerbated by the log scale in the y-axis) due to the model performance on a given dataset varying noticeably for different binary classification class pairings, which is reasonable given that certain class pairs may be harder to distinguish between over others, a variability that is heightened by our relatively low training set sizes. In any case, the error bars fall within the general test loss upper-bound region of the natural image results (distinctly lower losses than medical image models for a given $d_{data}$).
>
> > There are numerous similar examples of adversarial attacks natural imaging domain to the ones presented in Figure 4. What is special about these examples compared to the ones presented in even the earliest papers in this field (e.g., Szegedy et al. 2013, Intriguing properties of neural networks)?
>
> Good question; the adversarial examples shown in Fig. 4 are created with the same gradient-based technique (FGSM) of the early papers you describe (Goodfellow et al. 2014, and Szegedy et al. 2013, etc.) and they are not specially designed for any characteristic of medical images. As such, we have moved this figure to Appendix G as it doesn't show any unique visual behavior on medical images which doesn't also happen for attacked natural images, and so isn't necessary for the main text.
>
> Please proceed to the next comment for part 3 of our response ( to your very last question).

---

> ### Author Response · Authors · 2023-11-18
> **Response to Reviewer xVDy (Part 3)**
>
> > Can you provide some intuition on why in Figure 5 and for the VGG models, the representation intrinsic dimension doesn’t change that much for medical imaging domain?
>
> We speculate that because the VGG models have many more parameters than the ResNet models (133M - 144M compared to the ResNet models having 12M - 26M), they may not have fully fit to the datasets as well as the ResNets did (given our relatively small training set sizes), which could partially not fulfill assumptions we made (Sec 3, first 2 paragraphs) when deriving our scaling law (Eq. 4). If the models did not fully fit, then models trained on different datasets would be more similar to each other, resulting in the learned representation characteristics (intrinsic dimension) not changing much. This seems to be supported by this phenomena being more visible for the larger VGG models. We suspect that this under-fitting phenomena would fade if we used higher training set sizes, but this was not possible to do for some of our datasets (see footnote 3 on page 4).

---

> ### Comment · Reviewer_xVDy · 2023-11-23
> **Response to rebuttal**
>
> I'd like to thank the authors for their rebuttal and the updated results in the paper. After reading the rebuttal, I've increased the rating from 5 to 6.

---

### Author Response · Authors · 2023-11-18
**Author Responses + Revisions Summary**

Thank you to all reviewers for their helpful suggestions and valuable feedback. We will address each of your comments and questions point-by-point in our respective responses to you, and describe any corresponding additions or changes to the paper where applicable. Along with addressing all of your comments in our responses and/or by updating the main text, notably we have added new experiments and results in Appendices B.4, B.5, B.6, E.1, and F, which include a likelihood analysis of our proposed scaling law with respect to observed generalization data, evaluation of a new dataset in a third domain, further analyses of the behavior of dataset intrinsic dimension and label sharpness (e.g., with respect to image resolution), an example of a practical application of our findings, and others.

---

### Meta-Review · Area_Chair_E4YE · 2023-12-03

**Metareview:**

The paper investigates how neural networks learn from natural vs from medical images.
After the rebuttal, all reviewers are in agreement to accept the paper. The reviewers note that the paper addresses an important knowledge gap in understanding generalization, provides theoretical and empirical results, and is well written.
The reviewers also identified some weaknesses that to a large extent were addressed by the rebuttal. One weakness that remains was raised by reviewer fAwr: "A central claim of this paper is that "medical images are typically more challenging to generalize to". I don't think that this is a widely held belief or has been consistently shown throughout the literature". I agree with the reviewer that this claim is not accurate and overly broad, and suggest removing it as well as similar claims and overly broad statements.

**Justification For Why Not Higher Score:**

The claims are not sufficiently broadly validated to justify a higher score.

**Justification For Why Not Lower Score:**

The paper started an interesting discussion on differences of learning from different domains, which is useful and for interest for the community.

---

### Decision · Program_Chairs · 2024-01-16

Accept (poster)